

# Indications for a potential synchronization between the phase evolution of the Madden-Julian oscillation and the solar 27-day cycle

Christoph G. Hoffmann[1] and Christian von Savigny[1]

[1]Institute of Physics, University of Greifswald, Felix-Hausdorff-Str. 6, 17489 Greifswald, Germany

**Correspondence:** Christoph Hoffmann (christoph.hoffmann@uni-greifswald.de)

**Abstract.** The Madden-Julian oscillation (MJO) is a major source of intraseasonal variability in the troposphere. Recently, studies have indicated that also the solar 27-day variability could cause variability in the troposphere. Furthermore, it has been indicated that both sources could be linked, particularly, that the occurrence of strong MJO events could be modulated by the solar 27-day cycle.

In this paper, we analyze whether the temporal evolution of the MJO phases could also be linked to the solar 27-day cycle. We basically count the occurrences of particular MJO phases as a function of time lag after the solar 27-day extrema in about 38 years of MJO data. Furthermore, we develop a quantification approach to measure the strength of such a possible relationship. and use this to compare the behavior for different atmospheric conditions and different datasets, among others. The significance of the results is estimated based on different variants of the Monte Carlo approach, which are also compared.

We find indications for a synchronization between the MJO phase evolution and the solar 27-day cycle, which are most notable under certain conditions: MJO events with a strength greater than 0.5, during the easterly phase of the Quasi-biennial oscillation, and during boreal winter. The MJO appears to cycle through its 8 phases within 2 solar 27-day cycles. The phase relation between the MJO and the solar variation appears to be such that the MJO predominantly transitions from phase 8 to 1 or from phase 4 and 5 during the solar 27-day minimum. These results strongly depend on the used MJO index such that

the synchronization is most clearly seen when using univariate indices like OMI in the analysis, but can hardly be seen with multivariate indices like RMM. A weaker dependence of the results on the underlying solar proxy is also observed.

Although we think that these initial indications are already worth to be noted, we do not claim to unambiguously prove this relationship in the present study; neither in a statistical, nor in a causal sense. Instead, we challenge these initial findings ourselves in detail by varying underlying datasets and methods and critically discuss resulting open questions to lay a solid

foundation for further research.

## 1    Introduction

The solar electromagnetic radiation is the major energy source of the earth system. Although being usually described with the solar constant ($1361\,\mathrm{Wm^{-2}}$), the total solar irradiance (TSI) is subject to variations on different time scales with the most prominent one being the solar 11-year cycle. Whereas the variation of the TSI is only on the order of 0.1%, it differs among

the spectral regions and is particularly strong in the UV (e.g, Coddington et al., 2015, and references therein). Of interest

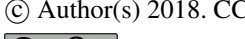



for the present study is the solar 27-day cycle, which is a combined result of the differential rotation of the sun and irradiance inhomogeneities on the solar disc. The amplitude of the 27-day cycle is generally smaller than that of the 11-year cycle, but can be on the order of 50% of the 11-year amplitude in the UV during strong events. The 27-day cycle is not perfectly periodic, but exhibits some variability, so that the 27 days have to be seen as a mean period of a quasi-periodic process. When using terms

like the "solar cycle", "solar maximum", etc., we always refer to the 27-day variations in this paper if not stated otherwise.

The solar variations introduce atmospheric variability and many effects have been identified in the past, particularly in the middle atmosphere, where the strongly varying UV is important. Signatures of the 27-day cycle have been found in, e.g., temperature (Hood, 1986; von Savigny et al., 2012; Thomas et al., 2015), trace gases (e.g., Hood, 1986; Robert et al., 2010; Thomas et al., 2015; Fytterer et al., 2015; Lednyts'kyy et al., 2017), polar mesospheric clouds (e.g., Robert et al., 2010;

Thurairajah et al., 2017; Köhnke et al., 2018), and very recently in radio wave reflection heights (von Savigny et al., 2018). The interactions between solar and atmospheric variability are still subject of ongoing research, which aim at both identifying more affected parameters and elucidating the underlying mechanisms.

In addition to implications in the middle atmosphere, a discussion of possible of 27-day signatures in the troposphere came up recently, mostly in the context of convection and clouds (Takahashi et al., 2010; Hong et al., 2011; Miyahara et al., 2017;

Hood, 2018), but also related to temperature (Hood, 2016). Even more than for the middle atmospheric effects, questions concerning the mechanisms behind the tropospheric signatures arise. Hood (2018) summarizes the two major classes of ideas; on the one hand the "bottom-up" mechanisms, which assume that the only slight variations of the TSI produce strong enough heating changes directly in the troposphere to generate the observed modulations in the upper troposphere. And on the other hand the "top-down" mechanisms, which consider the stratospheric effects of the stronger UV variations as starting point; via

a chain of effects the stratospheric changes could result in a change of upper tropospheric static stability and with that in a change of tropospheric deep convection with implications for clouds and temperature. Another mechanism, particularly for a connection between clouds and the solar variability, has been proposed in a few variants (e.g., Svensmark, 1998; Marsh and Svensmark, 2000), but has also been heavily criticized (e.g., Damon and Laut, 2011) and is mentioned here only for completeness. It considers a connection of cloud condensation nuclei and incoming galactic cosmic rays, whose flux is affected

by solar activity.

Independent of a possible solar influence, there is a known important source of tropospheric variability on the intraseasonal time scale, the Madden-Julian oscillation (MJO). It has first been reported by Madden and Julian (1972), more recent reviews of properties and implications are found in Zhang (2005) and Lau and Waliser (2012). In brief, it is a planetary-scale pattern in the tropics consisting of a region with anomalous strong deep convection flanked by two regions of weak deep convection to

the east and to the west. This pattern evolves over the Indian Ocean and travels eastward across the Maritime Continent until it decays in the Pacific. This temporal evolution is usually split into 8 phases as originally suggested by Madden and Julian (1972, Fig. 16). The MJO pattern reappears periodically, however, the period is strongly variable in a range between 30 and 100 days (Zhang, 2005). The MJO is the dominant component of intraseasonal variability in the tropical troposphere with strong influences on, e.g., rainfall and the genesis of tropical cyclones in the respective regions. In addition, there are also increasing

indications for an entanglement of the MJO in teleconnections and, hence, for an influence of the MJO in the extratropics



(e.g., Garfinkel et al., 2014). Due to its intraseasonal time scale and the large spatial scales, one important motivation for MJO research is that it could help to push the limits of weather forecasting skills towards longer periods (Zhang, 2013).

In addition to the tropospheric implications, also indications for interdependencies with the middle atmosphere have been brought up, particularly with ozone and temperature (e.g., Tian et al., 2007; Zhang et al., 2015; Yang et al., 2017). Of particular

interest for the present study is the finding that the MJO depends on the state of Quasi-biennial oscillation (QBO) (e.g., Son et al., 2016; Yoo and Son, 2016; Marshall et al., 2017). The QBO represents a quasi-periodic reversal of the stratospheric equatorial zonal winds with a mean period of 28 months (e.g., Baldwin et al., 2001). Briefly, it affects the MJO strength particularly during boreal winter such that the MJO is stronger during the QBO easterly phase.

In the context of solar-induced tropospheric variability there is a two-level interest in the MJO. First, it might be difficult

to distinguish both possible causes for intraseasonal tropospheric variability, since the MJO acts on time scales (starting with 30 d) close to the solar 27-day variation. Hence, suspected 27-day signatures in the troposphere might in reality be connected to the MJO. Second, in the light of recent publications, which are outlined below, it appears at least conceivable that the MJO is itself influenced by the 27-day cycle. From this point of view, the MJO might be a pathway for the 27-day solar signal into the troposphere. Hence, the three topics solar 27-day variation, MJO, and tropospheric variability on intraseasonal time scales

might be interconnected.

An example for rather the first level is the publication by Takahashi et al. (2010), which reports on 27-day variations found in the cloud amount over the Western Pacific region. These results are based on a frequency analysis of OLR data in place of direct cloud amount data. The authors are cautious with speculating on possible mechanisms, but also briefly mention that the spectral analysis shows indications of MJO activity and that some kind of interdependency cannot be ruled out.

The second level, a possible modulation of the MJO itself by the 27-day solar cycle, has been proposed by a series of studies (Hood, 2016, 2017, 2018). The study by Hood (2016) is actually focused on a tropospheric temperature response to solar 27-day variations. A modulation of the MJO is discussed as part of the mechanism, which brings the temperature signal into the troposphere. An initial investigation of this hypothesis shows a change of the occurrence of the particular MJO phases 1, 7, and 8 after solar 27-day extrema, which is considered to be consistent with the tropospheric temperature change. Hood (2017)

directly deals with a solar modulation of the MJO, but is focused on the solar 11-year cycle and the occurrence rate of strong MJO events instead of MJO phase occurrences. The study indicates that the MJO is influenced by solar 11-year variations during boreal winter. This influence is similarly important as the previously mentioned QBO modulation and might work with a similar mechanism: the modification of upper tropospheric stability. This also means that both influences have to work in the same direction (e.g., QBO easterly phase and solar minimum) to get a detectable MJO change. Hood (2018) also analyzes

the occurrence of strong MJO events, but returns to the solar 27-day variations. A statistical relationship between solar 27-day variations and the occurrence of strong MJO events is indeed found during the boreal winter and spring months from December to May. Particularly, strong MJO events (amplitudes greater than 2) are decreased following solar maxima and vice versa. As before, this effect is stronger under QBO east conditions.

The analysis presented here contributes to the critical examination of a possible linkage between the solar 27-day cycle

and the MJO. It is complementary to the previous studies, as it deals with the temporal MJO phase evolution instead of MJO





strength. Analyzing the temporal evolution focuses on a special aspect: the relation of the periods of both processes; first, the range of possible MJO periods starts close to the period of the solar 27-day cycle. And second, the mean periodicity of the MJO is with 50 to 60 days approximately twice that of the solar 27-day variability, which turns out to be of interest in the following. Overall, it is analyzed here if there are similarities and regularities in the temporal evolution of both processes and

we will show that a kind of coincident behavior can indeed be found in a statistical sense, which is partly surprisingly clear. However, we would like to emphasize that we do not try to prove a causal relationship between the solar 27-day cycle and the MJO phase evolution at this early stage. Likewise, we do not try to establish a particular mechanism. Instead, we aim at describing the statistical features of a combined inspection of both quasi-period processes as a basis for future research.

In Sect. 2 we describe the analyzed datasets and the initial filtering of the data. In Sect. 3 the basic analysis idea it outlined

first, before the essence of the found statistical relationship between solar 27-day cycle and MJO phase evolution is demonstrated based on a particularly clear example. In Sect. 4 questions concerning the generalizability of this example are addressed. For this a numerical approach to measure the strength of the relationship is developed first, before the analysis is applied to different selections of the underlying data. A discussion of major open questions and the conclusions are found Sect. 5.

## 2   Datasets and Filtering

Basically two pieces of information are needed to perform the present analysis: the time series of the solar activity and the MJO in the past.

The solar activity is represented by several proxy time series. We use primarily the Lyman alpha flux[1] (Woods et al., 2000) as indicator for solar activity. In addition, we have also performed the same analysis with the F10.7 radio flux index[2] (e.g., Tapping and Charrois, 1994, and references therein) and UV radiation at 205.5 nm simulated by the NRLSSI2 model[3] (Coddington et al.,

2015), as well as similar data of a previous model version.

Many different indices have been developed to compactly describe strength and phase of the MJO at a given time. These indices are usually calculated from either circulation data or information on cloudiness. The latter is usually represented by outgoing longwave radiation (OLR) data. Some approaches also combine both aspects to form multivariate indices (Straub, 2013). One of the latter indices is the Real-time Multivariate MJO index (RMM), which became the standard after its publi-

cation by Wheeler and Hendon (2004). A variant of RMM is the Velocity Potential MJO index (VPM) introduced by Ventrice et al. (2013), in which the OLR information is replaced by the a velocity potential. This leads to a better MJO representation during boreal summer among other advantages. More recently, Kiladis et al. (2014) introduced the OLR-based MJO index (OMI), which is a univariate index solely based on OLR data. It overcomes drawbacks of RMM (Straub, 2013; Kiladis et al., 2014) at the expense of the real-time capability. This disadvantage is, however, not of importance for retrospective analyses,

so that OMI has become an important index at least for these cases. Kiladis et al. (2014) also introduce a second univariate OLR-based index, the filtered MJO OLR index (FMO), which is easier to calculate than OMI. Kiladis et al. (2014) point out

---

[1]downloaded from http://lasp.colorado.edu/lisird/data/composite_lyman_alpha/

[2]downloaded from http://lasp.colorado.edu/lisird/data/penticton_radio_flux_observed/

[3]downloaded from http://lasp.colorado.edu/lisird/data/nrl2_ssi_P1D/



that all these different indices lead to similar results concerning the statistical gross features of the MJO, but differences are to be expected when working on the basis of individual MJO events.

Since our analysis does not depend on real-time information, we use primarily the OMI index[4]. An example of the OMI data as well as of the Lyman alpha solar proxy is shown in Fig. 1. Additionally, we have also applied our analysis to the RMM index[5], the VPM index[6], and the FMO index[7]. All these indices provide two coefficients each, which are transformed into MJO phase and strength by basically applying a transformation from Cartesian coordinates, in which the index coefficients are given, to polar coordinates. Radius and phase angle of the polar coordinates then correspond to MJO strength and phase, respectively. Note that there are different conventions among the different indices for the attribution of the index coefficients to the Cartesian coordinate system (Kiladis et al., 2014). The phase angle is then divided into 8 ranges of 45° each, which represent the 8 MJO phases mentioned before (e.g., Wheeler and Hendon, 2004).

The availability of the MJO indices is the limiting factor for the temporal extent of the analysis. The OMI index starts in 1979 and ends in August of 2017 at the time of the analysis, hence covers about 38 years. The other MJO indices cover roughly a similar period. All datasets are available with daily resolution so that the analysis is performed on a daily resolved grid.

As part of the analyses described in Sect. 3 and Sect. 4, the datasets are filtered with respect to geophysical properties: First, only days are considered, during which the MJO strength exceeds a particular threshold. Second, as the marker for the start of a new solar 27-day cycle, the solar minimum is used mostly, but the solar maximum can also be selected. Third, from the detected solar cycles, the relevant ones can be selected according to the season and, forth, they can be filtered according to the state of the QBO. For the latter, 50 hPa zonal wind data[8] from radiosondes in the tropics have been used (Naujokat, 1986). For the determination of the QBO phase, simply the sign of the wind data is used with positive values denoting the westerly phase and negative values denoting the easterly phase.

## 3   Essential nature of the potential relation between MJO phase evolution and solar 27-day cycle

As the basic analysis step, we check whether individual MJO phases appear preferentially at a particular state of the solar 27-day cycle. The idea is to count the number of occurrences of the individual MJO phases as a function of time lag after solar extrema. We analyze 28 days after each solar extremum; these temporal windows are called the epochs. This analysis is related to the approach of Hood (2016), but we treat all 8 MJO phases separately, while Hood (2016) focused on a combination of a few of them. We will demonstrate in the following that a preference for particular MJO phases depending on the solar 27-day state appears indeed to be present under certain conditions. We chose an experimental setup in this section for demonstration purposes, with which this relationship appears comparatively clear and we will discuss the ability to generalize these findings in Sect. 4.

---

[4]downloaded from https://www.esrl.noaa.gov/psd/mjo/mjoindex/omi.1x.txt
[5]downloaded from http://www.bom.gov.au/climate/mjo/graphics/rmm.74toRealtime.txt
[6]downloaded from https://www.esrl.noaa.gov/psd/mjo/mjoindex/vpm.1x.txt
[7]downloaded from https://www.esrl.noaa.gov/psd/mjo/mjoindex/fmo.1x.txt
[8]downloaded from http://www.geo.fu-berlin.de/met/ag/strat/produkte/qbo/qbo.dat



The following explanation of the analysis approach is also illustrated in Fig. 1. The analysis starts with identifying the solar 27-day minima in the Lyman alpha solar proxy time series (solar 27-day maxima are calculated likewise for other experimental setups). For this, the anomaly of the Lyman alpha time series is calculated by subtracting the smoothed time series (35-day moving average), which removes the variations greater than 35 days. Also the shorter term variations are removed from the

anomaly by smoothing it with a 5-day moving average. In the resulting proxy anomaly time series, the local extrema are identified. Only extrema, with anomaly values of at least $0.2 \times 10^{11} \frac{photons}{cm^2 s}$ above or below 0 are considered. This is a relative conservative filtering of extrema candidates; to make sure that only clear cases are considered in the analysis, we risk that some actual extrema are missed by the algorithm. In total, the algorithm finds 243 solar minima in the 38-year period, which means that about 6500 days out of the about 14000 days are covered with considered epochs. Not considered extrema belong

mostly to solar 11-year minimum phases and to periods, during which the 27-day signal is not very pronounced although the solar activity might generally be higher. From this set only solar minima are selected, which occurred during boreal winter (December, January, February) and during the QBO easterly phase. This results in a set of only remaining 26 epochs. However, the filter criteria in this example are among the most restrictive ones, so that the number of 26 samples is roughly a lower boundary for the sample size of the following experiments.

For all remaining solar minima days, we count how often each of the 8 MJO phases has occurred. A phase occurrence is only taken into account if the MJO strength exceeds the threshold of 1 in the current example, so that the sum of all phase occurrences is usually lower than the number of considered epochs (19 occurrences in this example). This is not only done for exactly those days with the solar minima, but it is repeated for all time lags between 1 and 28 full days after each solar minimum. This results in one curve for each of the 8 MJO phases describing the number of occurrences as function of time lag

after solar minimum. These curves are shown for the current example in Fig. 2.

Considering that the MJO shows a great variability and that also the solar 27-day cycle is only a quasi-periodic process, one would expect that these curves are basically constant with strong noise contributions. This would mean that each MJO phase occurs without any preference similarly often at each time lag after the solar minimum, which in turn means that the MJO phases evolve independently of the solar 27-day cycle. And the first overall impression of the functions in Fig. 2 might

apparently reflect this expected chaotic nature to a certain extent.

However, a closer look reveals some structure in the functions. First, each of the 8 curves exhibits a maximum at a particular time lag. The maxima are partly quite pronounced (e.g., for MJO phase 1) and partly somewhat broader (e.g., for MJO phase 7), but a kind of maximum is recognizable for each of the MJO phases. This indicates that the MJO phases occur preferentially at a certain time lag after the solar minimum. Second, the positions of the maxima reveal a specific ordering. Starting with the

maximum of MJO phase 1 at time lag 3 d, the maxima of the phases 2, 3, and 4 follow monotonically with increasing time lag. MJO phase 5 starts again with a low time lag of 8 d followed again monotonically by the phases 6, 7, and 8.

This structure is more clearly visualized in Fig. 3, where the time lags of the phase occurrence maxima are shown for each MJO phase. The two sequences of monotonically increasing time lags for the phases 1 to 4 and 5 to 8 are clearly visible and constitute a sawtooth-like pattern.





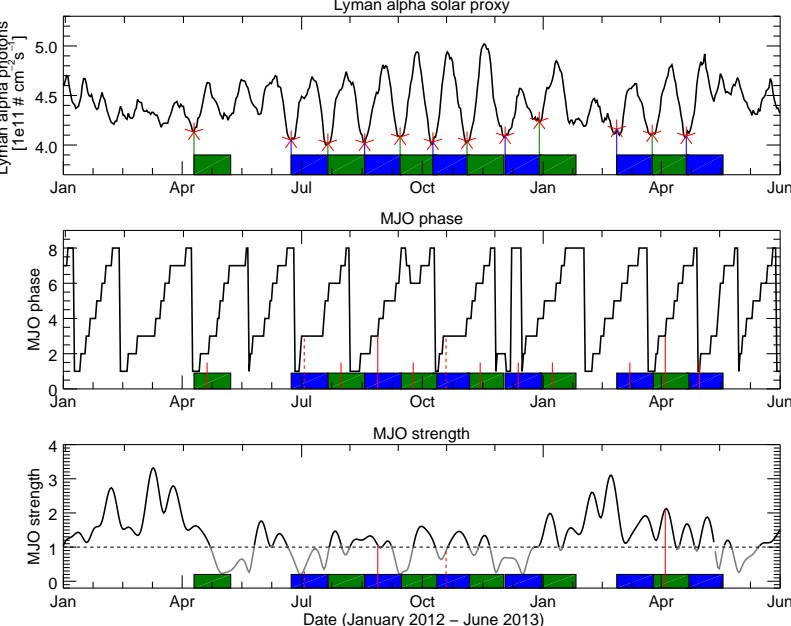

**Figure 1.** Examples of the analyzed time series for the period January 2012 to June 2013. The Lyman alpha solar proxy (top panel) shows pronounced 27-day variability particularly in the middle of this period. It is obvious that there is no perfect periodicity, instead the solar 27-day signal varies in strength and timing, so that it has to be considered as a quasi-periodic process. The same is true for the MJO phase evolution (middle panel; here represented by the OMI index). Overall, a periodic transition from phase 1 to phase 8 is clearly noticeable. However, the variable nature of the MJO is also evident; first from the deviations from the monotone phase ordering and second from the strong variations of the duration of the individual MJO cycles. Strong variability also appears in the corresponding evolution of the MJO strength (bottom panel, OMI index). The figure also indicates the basic steps of the analysis routine. Identified solar minima are marked with red stars in the top panel. The resulting 0 to 28 d time lag epochs are indicated by green and blue shaded areas in all three panels. The analysis then counts how often each MJO phase occurred for each time lag after solar minimum, e.g., how often MJO phase 3 occurred 10 days after solar minimum. To illustrate this example, all 10-day time lags are marked (all vertical red lines in the middle panel). During 4 of these days, the MJO was in phase 3 (longer solid and dashed red lines). However, finally only days are considered, during which the MJO strength exceeded a particular threshold (here set to 1, horizontal dashed line in the bottom panel). Considering this, the analysis results in 2 occurrences of phase 3 for time lag 10 (longer solid vertical red lines). This is repeated for all MJO phases and all time lags.

The appearance of this clear pattern is the major qualitative result of this study and the essential characterization of the possible relationship between solar 27-day cycle and MJO phase evolution. We think, that this result is quite remarkable considering that the MJO, although showing some kind of periodicity, is a highly variable phenomenon.

Based on the clearness of this pattern, it appears attractive to directly assume a causal synchronizing mechanism between the solar 27-day cycle and the MJO phase evolution, which would, however, be premature. Nevertheless, the mere appearance of this sawtooth pattern has at least two requirements. First, the mean period of the MJO should be twice as large as the mean





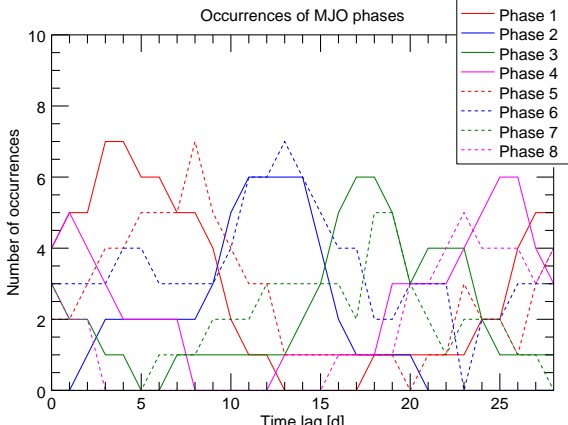

**Figure 2.** Number of occurrences of each MJO phase (one line per phase) as a function of time lag after solar 27-day minima. The figure shows a particular example: only solar minima during boreal winter and during a QBO easterly phase have been considered. After this filtering 26 epochs remain in the analysis. Furthermore, the MJO strength on individual days has to be greater than 1. The state of the MJO is characterized by the OMI index, the Lyman alpha solar proxy has been used to determine the solar minima.

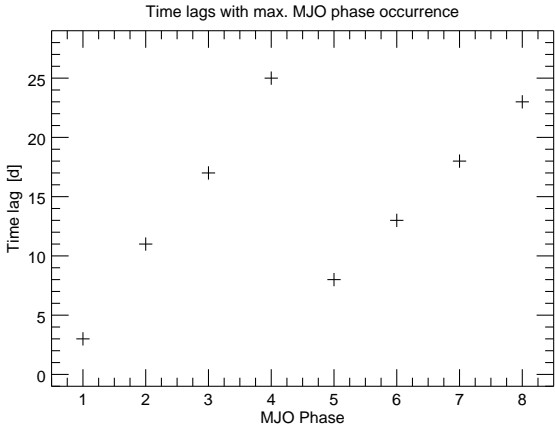

**Figure 3.** Position of the maximum occurrence numbers (measured in time lag after solar minimum) for each MJO phase. Datasets and filtering conditions are similar to those in Fig. 2.

period of the solar 27-day variation. Hence, it should be about 54 d, which is well in the already known range of periods. Taking into account, though, that the instantaneous MJO period varies strongly, the second requirement is needed, namely that there should be a predominant phase relation of the solar 27-day variations and the MJO phase evolution during the complete analyzed period, i.e. the MJO is predominantly around phase 1 or around phase 5 at solar minimum. Otherwise, the sawtooth shape would be arbitrarily shifted over the MJO phases for certain sub-periods, so that the pattern would finally be averaged out when taking the complete analyzed period into account. These requirements are obviously to a large extent fulfilled in





the present example, however, the question remains whether this fact really demands for a causal mechanism or if it could also be a coincidence in the analyzed period. Furthermore, a possible causal mechanism would have to explain why the solar 27-day variation produces a variation with a doubled period, i.e. why there are two possible MJO phases at each solar state. We emphasize again that it is not our aim to prove such a causal connection in this study. Instead, we aim at carving out more

statistical characteristics of this connection from the dataset itself in Sect. 4 as a first step. This helps to get a clearer picture under which conditions such a connection might exist.

In the light of the present findings, it is in order to briefly comment on some results in Hood (2016), which are also based on counting the occurrences of MJO phases as a function of time lag after solar 27-day maxima or minima. However, in contrast to the present study, the MJO phases are not treated individually, but only the cumulative occurrence of MJO phases 1, 7, and

8 is evaluated, which is motivated by the particular questions in this analysis. The author finds that the cumulative occurrence of these phases is enhanced in the days after solar minimum and reduced about 10 days after the solar minimum. Having the present results in mind, it does not seem to be a very reasonable choice to combine the particular phases 1, 7, and 8 as their positions of maximum occurrence represent 3 (of possible 4) different time lag ranges (Fig. 3). Instead, if one wants to group the phases with respect to the solar cycle, it would be more plausible to overlay the two lines of the sawtooth pattern, which

means that the following pairs of MJO phases belong together in their relation to the solar 27-day cycle: 1 and 5, 2 and 6, 3 and 7, 4 and 8. Additionally, it should not be expected that the phase package 1, 7, and 8 behaves contrarily to the "opposite" phase package, consisting of the MJO phases 3, 4, and 5 (which Hood (2016) does not claim, but what the reader might intuitively think). Instead this package represents similar maximum occurrence time lags as the first package 1, 7, 8 (Fig. 3) and should behave similarly. Having this in mind, the conclusions drawn based on these results in Hood (2016) should be reconsidered,

especially because the author has pointed out the initial character of these results himself.

## 4   Quantitative examination of the potential relation between MJO and solar cycle for different conditions

It is our aim to challenge the hypothesis of a relationship between the solar 27-day cycle and the MJO phase evolution by diversifying the setups of the numerical experiments. That means that the same analysis is repeated for different choices of atmospheric conditions, underlying datasets, and also implementation details. To do so, a quantity is needed first that measures

the strength of the relationship and, hence, makes the results for different setups comparable.

### 4.1   Brief description of the quantification approach

Based on Fig. 3, it is intuitive to define such a quantity as the similarity of the pattern constituted by the 8 data points to a sawtooth function. Numerically, this similarity can be estimated by fitting a sawtooth function to the data points. The goodness of fit $\chi^2$, which basically sums up the quadratic deviations between data points and the fitted sawtooth function, could then

be a natural choice for such a measure; the smaller the value of $\chi^2$, the better is the similarity to a sawtooth function and the stronger in the relationship between solar 27-day cycle and MJO evolution.





However, the common definition of $\chi^2$ has to be modified in two aspects to be a suitable measure in the present context. This is described in detail in the Appendix A and mentioned here only briefly: First, the calculation of the individual deviations has to account for the fact that the time lags are periodic with a periodicity of 27 d. This is considered in the quantity $\chi^2_{per}$ defined in the Appendix A2. Second, the common weighting of each data point with its reverse variance $\frac{1}{\sigma_i^2}$ works in the direction

that a higher uncertainty (greater standard deviation $\sigma_i$) leads to a smaller $\chi^2$. This is useful for the numerical fitting routine, but works in the wrong direction for the present application, the measurement of deviations. For this application, a higher uncertainty should result in a greater value of the deviation, which reflects a weaker certainty of the found relationship. Both aspects are considered in the quantity $X$, which we defined as the measure of the deviation in the Appendix A3. This quantity $X$, simply called "deviation" in the following, is the used measure for the strength of the relationship in this study; a lower

deviation indicates a stronger relationship between solar 27-day cycle and the MJO phase evolution.

Altogether, our analysis routine comprises the following steps:

1. Performing the analysis steps described in Sect. 2 and 3

    (a) Identification of the solar extrema dates.

    (b) Filtering of the input data according to the experimental setup.

(c) Counting of the occurrences of the individual MJO phases as a function of time lag after the solar extrema.

    (d) Identification of the time lags with maximum occurrence number for each MJO phase.

2. Estimation of the uncertainty of the derived time lags using a bootstrap method (in more detail described in Appendix A4)

3. Fitting the sawtooth function to the derived time lags of maximum occurrence for the 8 MJO phases using the previously calculated bootstrap uncertainties as weights. As mentioned before, the fit is performed under consideration of the 27-day

periodicity of the time lags, hence by minimizing $\chi^2_{per}$ instead of $\chi^2$. For the same reason, we have fixed the amplitude of the sawtooth function in the fit to a value of 27 d. Assuming, based on the previous results, that the mean periodicity of the MJO is with 54 d twice the mean periodicity of the solar 27-day cycle, we have also fixed the period to 4 MJO phases (only half of the 8 MJO phases are experienced during one solar 27-day cycle). The only free parameter of the fit is the phase $\phi_{St}$ of the sawtooth function. As the fitting routine might not directly find the global minimum of $\chi^2_{per}$ and,

hence, the result might depend on the first guess of $\phi_{St}$, the fitting procedure is repeated with the first guesses of $\phi_{St}$ systematically varied between 1 and 8. The result with the minimal $\chi^2_{per}$ is then further on considered.

4. Calculation of the measure of deviation $X$ between data and fit using the bootstrap uncertainties as weights. Although the measure of the "deviation" is the direct quantitative result of the analysis, we are conceptually interested in the opposite, the "similarity" of the pattern in the data and the sawtooth function. In the following, we will use both terms equally

in the sense that a small deviation means high similarity, which in turn means a stronger relation between solar 27-day cycle and the MJO phase evolution. Furthermore, we will not put emphasis on the physical units of $X$, which depend on the weighting factors. Since only the variations in the results are of interest and not the absolute values, we will simply assume that $X$ is given in arbitrary units.





5. Estimation of the significance of the quantified relationship. For this, the probability $p$ that the value of the deviation $X$ could be the product of only random features in the data is calculated. This is achieved with a Monte Carlo (MC) approach, which means that the input data is repeatedly modified with random numbers and the complete analysis procedure is applied to a large number (1000) of such randomly modified input data representations. The probability $p$ is then simply calculated as the percentage of the random experiments, which resulted in a lower deviation measure. This probability value then used to quantify the significance of the respective result; the lower the probability $p$ that a low deviation can be reproduced with random numbers, the higher is the significance of the result. We have implemented different possibilities for the creation of the random data. These are outlined together with the discussion of the respective results in Sect. 4.5. We use as standard method in the following the most conservative implementation, i.e., the one, which indicates most rarely significance of the results. This method is based on randomly shifting the original solar extrema dates by up to $\pm 6$ days and is also explained in more detail in Sect. 4.5.

An example of the fitting process, which corresponds to the case previously discussed in Sect. 3, is shown in Fig. 4. More examples are included in the supplement. After having performed this routine, a measure of the deviation between the sawtooth pattern in the empirical data and an analytical sawtooth function is known together with the fitted phase. This deviation characterizes the strength of a possible systematic relationship between the solar 27-day cycle and the MJO phase evolution. In the following, we will apply this approach to a variety of different experimental setups, which can then be compared among each other.

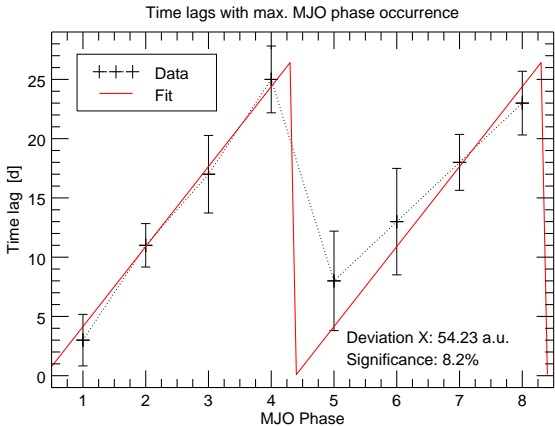

**Figure 4.** Time lags of the maximum occurrence for each MJO phase as in Fig. 3, but including results from the quantification approach: an estimation of the time lag uncertainties, a fitted sawtooth function, the fit deviation $X$, and a significance estimation.



## 4.2 Influence of the numerical setup

Before we discuss the results in detail, we note that the analysis is, like most others, subject to well justified but strictly speaking arbitrary choices. Wherever possible, we have repeated the analysis with different realizations of these choices and have convinced ourselves that our main conclusions do not depend on these choices.

One choice is the definition of the epoch period. We have defined an epoch to start with a solar extremum and then last for 28 days. This is a natural choice, since -if any relation can be substantiated- we expect the sun to be the driver of the MJO phase evolution so that it makes sense to study the atmospheric response in the period after the solar extremum. However, a possible mechanism does not guaranty a direct response of the atmosphere in the following 28 days. Instead the response could manifest itself also during the solar cycles afterwards. Hence, no unambiguous starting point of an epoch can be fixed and it

would also be possible to, e.g., center the solar extremum in the epoch period, so that it covers the time lags from -14 days to 14 days, like it is done in many studies. Interestingly, we found that some of our conclusions appear even clearer using this alternative choice of the epoch windows. Currently, we cannot decide whether this is a feature of the studied relationship, or if it is a random effect. Therefore, we decided to include the more conservative option of the 0 to 28 day epoch into the paper, but show the alternative results in the supplement.

Another choice is the use of squared weights in the definition of the deviation $X$ as mentioned in Sect. A3. Hence, we have also repeated the calculations with constant weights (we have chosen $w_i = 1/8$, so that the sum over all 8 weights is unity), so that all data points are weighted with the same factor. Although these results are not interesting from an atmospheric point of view, we have also included them in the supplement, to convince the reader that the conclusions are not influenced by the definition. However, for the interpretation of these alternative calculations, one has to note that the significance analysis cannot

lead to very realistic results in the case of these arbitrary constant weights; since the values of these weights directly influence the value of $X$, the choice of weights directly influences the probability to gain a higher or lower $X$ based on a random dataset. And whereas the original calculation of the weights considers the real spread of the data, leading to weights, which actually characterize the dataset, the constant weights are completely unconnected with the dataset. What can still be seen from the results with constant weights is that the qualitative comparison of results with different experimental setups is similar and,

hence, the conclusions are not dominated by the kind of weighting.

As an example, we have also included the results of both alternative calculations (the centered epoch definition as well as the constant weighting) in the presentation of the first experiment (Fig. 5, which is discussed in Sect. 4.3). Afterwards, all results will be based on the 0 to 28-day epoch and the squared bootstrap uncertainties as weights.

## 4.3 Influence of atmospheric conditions

In the following experiments, one parameter of the analysis will be varied, while the others are kept constant with specific values. We used indications from pretests and previous studies, to choose standard values for the non-varied parameters, which lead to the clearest results and, hence, allow the best conclusions concerning the particular influence of the varied parameter. For the overall conclusions, the values of all filters have finally, of course, to be considered at the same time. An example



**Table 1.** Parameters varied to quantify their influence on the possible relation between solar 27-day cycle and MJO phase evolution.

| Parameter | Possible values | Standard value |
|---|---|---|
| Minimum MJO strength | 0...2.5 | fully resolved in most experiments, otherwise 1 |
| QBO phase | easterly, westerly, no filtering | easterly |
| Season | boreal winter (DJF), boreal winter and spring (DJFMAM), boreal summer (JJA), no filtering | boreal winter |
| Solar epoch trigger | 27-day maxima or minima | minima |
| MJO index | OMI, RMM, FMO, VPM | OMI |
| Solar proxy | Lyman alpha, F10.7, NRLSSI2 205 nm, NRLSSI1 205 nm | Lyman alpha |

of more relaxed filtering conditions is shown afterwards in Sect. 4.3.5. An overview of the varied parameters including the standard values in this and the following section (4.4) is given in Table 1.

### 4.3.1 MJO strength threshold

One major parameter for all MJO studies is the minimum MJO strength, which has to be reached, for an MJO event to be
considered. Very often, a value of 1 is used, sometimes also a value of 2. We have examined in more detail the influence of this threshold on our results first. For this we have varied the MJO strength threshold between 0 and 2.5 in 0.1 steps. The other filter criteria correspond to the standard of the example in Sect. 3. The results are shown in Fig. 5, green line, for the standard numerical setup.

The results show comparatively low deviations, i.e., stronger indications for a connection between the solar 27-day cycle
and the MJO phase evolution, between thresholds of 0.8 and 2.1. In this center range, most of the results are significant at least at the 10% level, many at the 5% level and some at the 1% level. This means that the chance to derive lower deviations with randomly modified solar extremum dates is below this conservative estimate (see Sect. 4.1). Stronger deviations are evident at both edges, which is the expected behavior. First, stronger deviations for high MJO thresholds are directly caused by the low number of samples which remain (the sample size starts with 26 for MJO strength threshold 0, decreases to 19 for threshold 1,
which corresponds to the example in Sect. 3, and decreases further down to 1 in the present case with other restrictive filters for the threshold 2.5). Second, the stronger deviations for low MJO thresholds are caused by the consideration of periods, during which the MJO pattern (and with that the value of the MJO phase) can hardly be identified and the analysis incorporates mostly atmospheric variability not connected to the MJO.

We treat the deviation $X$ between data and fit as the main outcome of our analysis. Nevertheless, we also get values for the
phase $\phi_{St}$, which is the free fit parameter. It represents the phase of the MJO at the time of the solar extremum. Using the solar minimum as the trigger, we get a value for $\phi_{St}$ of about 0.3. This means that the MJO predominantly transitions from phase 8 to phase 1 or from phase 4 to phase 5 during solar minimum (compare the fitted sawtooth function in Fig. 4, particularly where it





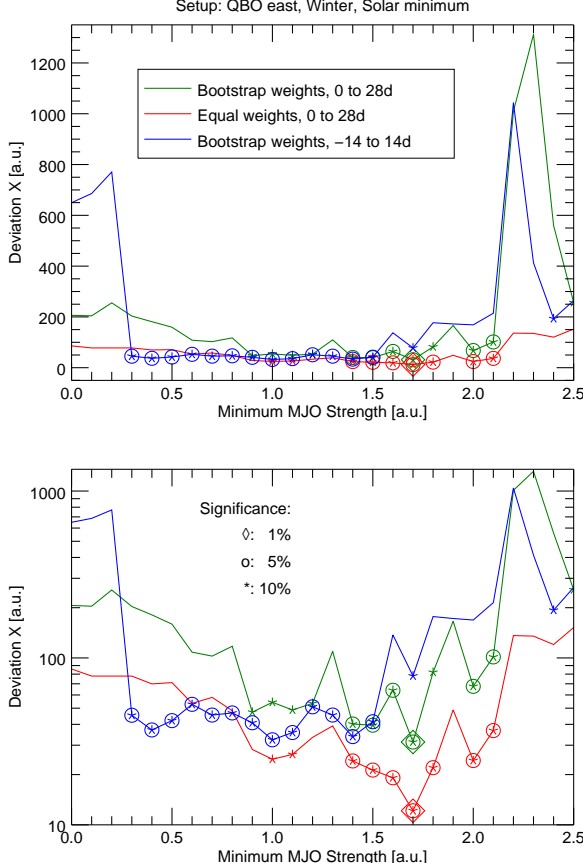

**Figure 5.** Calculated deviations $X$ between the empirical time lags to the solar trigger, at which the individual MJO phases occur maximally, and a fitted sawtooth function. The same data is shown in linear scaling (upper panel) for an overall visual impression and in logarithmic scaling (lower panel) for inspection of the smaller variations. According to our analysis approach, a lower deviation represents a stronger statistical connection between the solar 27-day cycle and the MJO phase evolution. Focus of this experiment is the dependence of the deviation on the MJO strength threshold. The green line shows these results for the standard numerical setup (epoch time lags from 0 to 28 d and squared weights from the bootstrap approach). Three levels of significance in the sense of the MC experiment are shown: 10%, 5%, or 1% chance of getting lower deviations with randomly modified solar extremum dates. The other two lines give an impression of the reaction of the results when the numerical setup is changed to, first, a different epoch definition (blue line) or, second, the use of constant weights (red line, see text for details).

approaches time lags of 0 days). Consistently, we find values for $\phi_{St}$ of about 2.2 when we use the solar maximum as a trigger, hence a shift by 2 MJO phases, which is a half of the sawtooth period, as expected. This means that the MJO predominantly transitions from phase 2 to phase 3 or from phase 6 to phase 7 during solar maximum. These numbers for the fitted phase are quite stable among the different MJO thresholds in this experiment, but also among the following experiments, whenever a

5    strong relationship between the solar 27-day cycle and the MJO evolution is found. This stability of the fitted phase among the





experiments is remarkable, as it also supports some kind of synchronization between the solar cycle and the MJO evolution in contrast to a hypothetical situation, in which the fitted phase strongly jumps depending on the particular experimental setup.

As mentioned before, Fig. 5 also shows the same results derived with slightly changed numerical setups as described in Sect. 4.2. First, for the alternative epoch definition (blue line), it is seen that the significant range of low deviations is somewhat

shifted to lower MJO strength thresholds and shows a bit less variability. Second, the deviations calculated with constant weights are generally lower, which is, however, due to the fact that the arbitrarily selected weights directly influence the value of the deviation $X$, so that a comparison of absolute values of $X$ is not reasonable. Only the variability within the red curve can be compared to the variability in the other curves and this looks quite similar. Overall, there are differences between the numerical setups, but the observation that relatively low deviations are found in the center of the MJO threshold range and

higher deviations at the edges is valid for all curves. In this sense, also the following conclusions will be independent of the numerical setup, so that only the results derived with the first setup will be shown in detail.

In the following we will present the results for the other numerical experiments in a similar way. However, we will mostly show the results only on the linear scale. That is because reading precise numbers of the deviations is not really important on this arbitrary deviation scale. Instead, the figures serve more for a visual comparison of the different experiments, which is in

our opinion easier with the linear scale in most cases.

### 4.3.2   Phase of the QBO

Yoo and Son (2016) showed that the MJO strength is influenced by the QBO in a way that the MJO is stronger during the QBO easterly phase. Hood (2017) suggests that the solar influence (in this case of the 11-year cycle) on the MJO activity might be masked by the QBO influence if both work in opposite directions, so that the MJO activity is strongest during solar

minimum and QBO easterly conditions. Hood (2018) finds that the influence of the solar 27-day variations on MJO strength is also strongest for the QBO easterly phase.

We have also checked the influence of the QBO in the context of the MJO phase evolution. For this, we excluded all epochs from the analysis, which do not match the wanted QBO phase and repeated the analysis, again resolved for different MJO strength thresholds. This has been done for boreal winter and a solar minimum epoch trigger. The results (Fig. 6) confirm a

strong influence on the relationship between MJO phase evolution and solar 27-day cycle, which is consistent with the previous studies. For the QBO easterly phase, we find relatively low deviations $X$ and significance levels between 1% and 10% in the center range of MJO thresholds. The deviations for QBO westerly periods are mostly more than one order of magnitude higher and the significance of all data points is worse than 10%. This means that there is no significant relationship between the solar 27-day cycle and the MJO evolution based of the sawtooth-fitting approach for QBO westerly phases in contrast to QBO

easterly phases. If no QBO filtering is applied, the deviations are, as expected, mostly between those of the QBO easterly and westerly filtering. Almost no data points are significant for this case.

Hence, we conclude that a possible relationship between the solar 27-day cycle and the MJO phase evolution is detectable only for QBO easterly conditions.





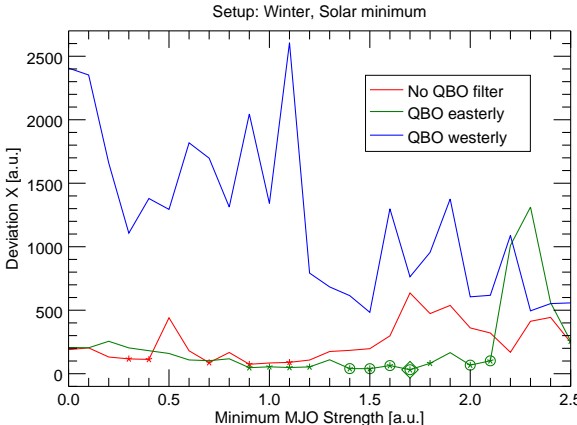

**Figure 6.** As Fig. 5, but with results resolved for the QBO phase. The green line corresponds directly to the green line in Fig. 5.

### 4.3.3 Seasons

It has been found before, that the MJO strength modulation by both the QBO and solar influences is mostly detectable during boreal winter, i.e., the during the months December, January, and February (Yoo and Son, 2016; Hood, 2017), sometimes extended by the months March, April, and Mai (Hood, 2018).

We have checked the seasonality in the context of the MJO phase evolution by restricting the considered epochs to the respective months. Indeed, also our results (Fig. 7) show that a strong relation between the solar cycle and the MJO phase evolution is indicated predominantly for boreal winter. A similarly strong relation is also seen for boreal winter extended by the spring months, which we analyzed for the sake of comparability to Hood (2018). During boreal summer, the deviations are partly an order of magnitude higher and only rarely significant. Data for boreal autumn has not been computed for reasons of

computation time. The unfiltered, i.e., year-round data leads to deviations, which are mostly located between the extremes and only rarely significant.

We have to note that the findings differ in this case somewhat among the alternative numerical setups (see Sect. 4.2). Particularly that boreal winter extended by spring behaves similar to winter only is not true for the numerical setup, in which the epoch covers -14 to 14 days around the solar extremum (see Fig. S2 in the supplement). In this case only the boreal winter

data shows a clear, significant relationship, but not the data extended by spring. Although our results appear to be largely consistent with Hood (2018), this detail is not consistent, as Hood (2018) is also based on the centered epochs.

We conclude here that a possible relationship between the solar 27-day cycle and the MJO phase evolution is detectable only during boreal winter, although an extension into spring might be possible. The reasons for the seasonality are speculative, but likely connected to the reasons for the seasonality identified by Hood (2018) and maybe also to that of the QBO influence

identified by Yoo and Son (2016); particularly the seasonality of the MJO itself or the seasonality of the stratospheric residual circulation.





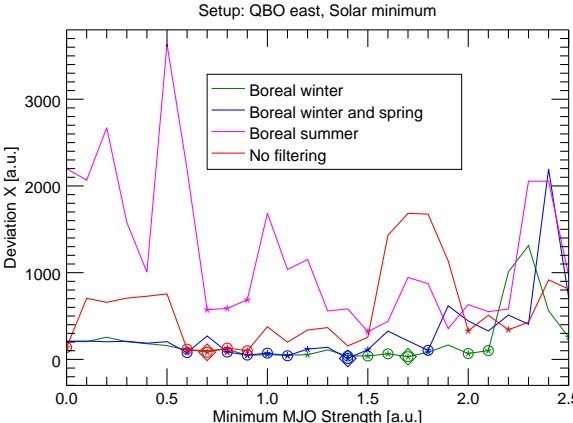

**Figure 7.** As Fig. 5, but with seasonally resolved results. The green line corresponds directly to the green line in Fig. 5. Boreal winter comprises the months December, January, and February, boreal spring the months March, April, and Mai, and boreal summer the month June, July, and August.

### 4.3.4 Solar minimum or maximum as epoch trigger

We have also checked whether it makes a difference to start the epochs with the solar 27-day minimum or maximum. It turns out that, in the reasonable range of MJO strength thresholds, the deviations $X$ are on a similar order of magnitude for both cases (Fig. 8) and, hence, that the choice of the trigger has no pronounced effect.

Nevertheless, looking more closely, one may note that the deviations are mostly at least a bit lower and more data points are significant when the solar minimum trigger is used. These differences should not be overinterpreted, but we would like to at least mention them, because they become more pronounced when the alternative experimental setup with centered epochs is used (Fig. S3 in the supplement). In this case, almost no data points are significant using the solar maximum trigger, whereas a continuous range over 13 data points is significant at the 5% level for the solar minimum trigger. Overall, the influence

of the trigger must therefore remain unclear in the present study. However, if a difference between both triggers could be substantiated in future, it could hint to possible mechanisms of a synchronization between the solar 27-day cycle and the MJO; it could indicate that the solar minimum is the actual trigger, which privileges certain MJO phases and that the MJO phase evolution runs freely afterwards, so that the results are more noisy when the analysis is started half of a cycle later using the solar maximum trigger. Also the observation that the MJO is predominantly before phase 1 during solar minimum would

appear consistent in this context (Sect. 4.3.1).

### 4.3.5 Relaxed atmospheric filter criteria

The previously presented experiments were designed such that one parameter was varied, while the other parameters were set to the optimal values. This, of course, limits the scope of the conclusions, since a clear relation between the solar 27-day cycle



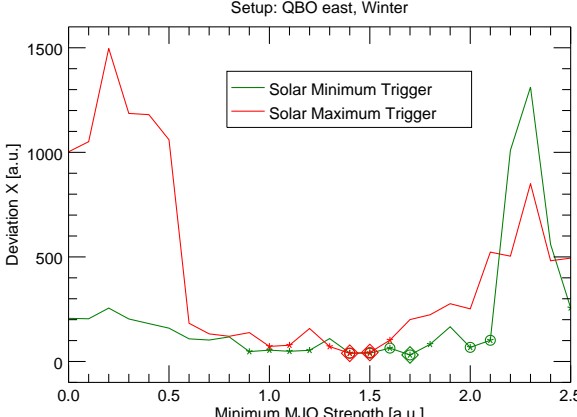

**Figure 8.** As Fig. 5, but also showing the analysis results for the epochs being started with solar maximum. The green line corresponds directly to the green line in Fig. 5.

and the MJO phase evolution is only indicated, when all conditions are met simultaneously, namely that the MJO strength threshold is in a range around 1, the QBO is in easterly phase, and the season is boreal winter. The previously shown negative results for other filter setups demonstrate that no relationship, which is significant at least at the 10%-level, is to be expected when one or more filter parameters are relaxed. However, as stated before, we have applied a quite conservative quantification

approach in terms of the selected numerical approach (Sect. 4.2) and the MC significance estimation (Sect. 4.1 and Sect. 4.5), so that it is still worth looking at an example with relaxed filter criteria to get an impression.

A second reason for looking into this example is that it overcomes one major drawback of the previous experiment design, namely that the number of samples is relatively low. Because of these low numbers one may wonder if the found relationship is a particular feature of exactly this sample, even if this risk is actually quantified by the MC analysis. In any case, it is

worthwhile to get an impression of results including more epochs.

As an example, the analysis has been repeated without the QBO- and without the season constraint. The derived time lags of maximum MJO phase occurrence are shown in Fig. 9, comparably with Fig. 4. The MJO strength threshold is set to 1, as in many other studies, but the results are comparable for similar MJO thresholds. With these criteria, about 140 of possible 243 epochs are considered instead of a few tens. As expected, the deviation $X$ is higher than in the optimally filtered case

(Fig. 4) and is not considered significant anymore with the probability to derive a lower deviation with random numbers being about 15%. But still the data points do not appear completely disordered. Instead they remind the eye still of the sawtooth-like structure analyzed before.

On the one hand, this could indicate that the described relation may also be there under different atmospheric conditions, but is superimposed by different kinds of variability. On the other hand, it may mean that the relation is so pronounced for specific

atmospheric conditions that the signature remains, even when other periods are included. In conclusion, although we analyzed already 38 years of data, the period does not seem to be long enough to significantly prove a statistical connection between




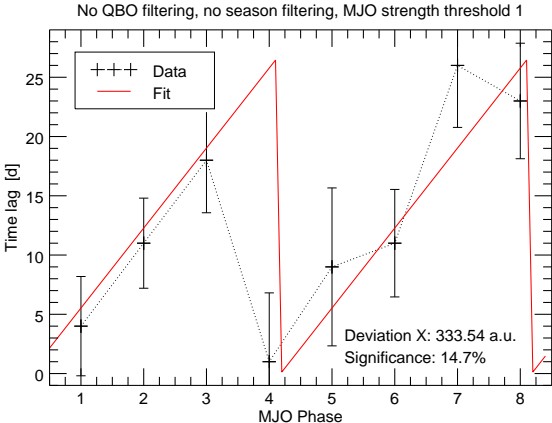

**Figure 9.** As Fig. 4, but showing results of an experiment without filtering for QBO phase and season. As described in the text, the results are not significant anymore on a level better than 10%. However, it is visually seen in this figure, that there are still indications for the sawtooth-like pattern instead of a totally unstructured behavior.

the solar 27-day cycle and the MJO phase evolution for more general atmospheric conditions than those described above, particularly not using our conservative MC approach. However, this does not necessarily mean that the relation is actually restricted to those conditions. Both, a longer dataset and refining the analysis approach to be less conservative, of course while remaining scientifically strict, could help to answer this in future.

5    Note that some more fit examples, which correspond to some cases of particularly strong deviations in the previous experiments, are shown in the supplement. Also there, rudimentary indications of a sawtooth structure can sometimes still be recognized although being highly insignificant.

## 4.4  Influence of underlying datasets

### 4.4.1  Influence of the MJO index

10  To describe strength and phase evolution of the MJO, several independent indices have been developed in the past (Sect. 2). In addition to the analysis based on the OMI index presented before, we have repeated the analysis with other important indices, particularly the major ones discussed in Kiladis et al. (2014): RMM, VPM, and FMO.

We have recalculated, e.g., the analysis described in Sect. 4.3.1, that is for the standard conditions boreal winter, QBO easterly phase, and solar minimum trigger. The results (Fig. 10) show clear differences between two pairs of the indices; 15  whereas the results for OMI and FMO show relatively low deviations $X$, which are largely significant at the 5% level, the other two indices RMM and VPM show much larger deviations, which are rarely significant. Hence, a relationship between the solar 27-day cycle and the MJO phase evolution is only indicated by OMI and FMO. While this result is somewhat surprising, the grouping of the indices appears plausible, since the pairs belong also conceptually together, as they are either the univariate



indices OMI and FMO based only on OLR data or the multivariate indices RMM and VPM, which also include circulation data.

This can be interpreted in two ways. First, it could mean that a potential connection of solar 27-day activity is not fully represented in the circulation-based indices RMM and VPM. This appears plausible, because OMI has a more precise repre-
sentation of the convective center and, hence, might better represent such subtle features we are looking for. This was exactly the reason for using OMI as the primary index as also done in, e.g., Hood (2017, 2018). But second, it could mean that we strictly did not identify a connection between the solar 27-day activity and the MJO but only between the solar 27-day activity and OLR. The signature would than appear more or less accidentally in the MJO indices and we would have found a similar solar variability-OLR connection as has been reported by Takahashi et al. (2010) using different methods. In this respect, it is in order to mention that Takahashi et al. (2010) and the OMI/FMO description paper by Kiladis et al. (2014) refer to the same OLR data basis, namely Liebmann and Smith (1996). If this second interpretation were true, it would mean that we have so far actually described the properties of the solar influence on OLR and not directly on the MJO. Although this was not our original objective, the results would still be interesting, as they underline the possibility of solar 27-day influences on the tropospheric parameter OLR. As in Takahashi et al. (2010), the remaining open question of interest would concern the mechanism of such a sun-OLR connection. An involvement of the MJO in such a mechanism would still be likely, as the OLR is, of course, influenced by the MJO.

Indeed, the fact that the properties of the relationship described so far are largely consistent with other MJO-related studies suggests that the MJO is at least involved in these interactions. Therefore, we propose to treat both interpretations equally serious for the time being. To be able to distinguish both interpretations, future research should further examine the solar influence on the data ingredients of the individual MJO indices and identify processing steps in the computation of RMM and VPM, during which the solar influence could get lost.

In conclusion, although the overall picture of the present results suggests that the MJO is actually somehow involved, the present study must strictly speaking remain inconclusive regarding the question, whether really the MJO is influenced by the solar 27-day cycle or if only OLR is affected or if maybe the OLR signal is generated by a modulation of the MJO. What can be stated, however, is that studies dealing with such subtle features of the MJO should repeat the analyses with different MJO indices and not arbitrarily select only one of them. This concerns also the mentioned series of papers on the solar influence on the MJO, which started with the RMM index (Hood, 2016), switched to OMI while mentioning RMM results (Hood, 2017), before relying complete on OMI (Hood, 2018). In the light of the current findings it would be of interest, whether the results of Hood (2018) are reproducible also with RMM or not.

### 4.4.2 Influence of the solar proxy

We have also checked the influence of the used solar proxy in our study. In addition to our standard proxy Lyman alpha, we have also included the F10.7 radio flux and for comparability with Hood (2016, 2017, 2018) the UV radiation data from NRL SSI models (Sect. 2).




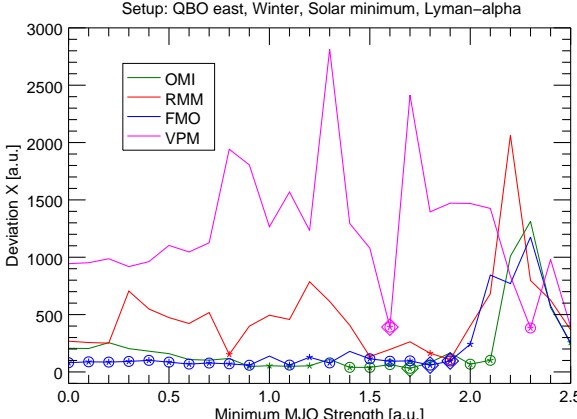

**Figure 10.** As Fig. 5, but also showing the analysis results for alternative MJO indices. The green line corresponds directly to the green line in Fig. 5.

Generally, the solar proxy data is not as fundamental as the MJO index for our analysis, since it is only used to generate a list of dates with solar extrema, which define the epochs. The algorithm to find these local extrema in the proxy time series (Sect. 3) depends on thresholds, which are adjusted for the particular proxy. Variations in the data among the proxies and the definition of these thresholds may cause the resulting list of extrema to be a bit different depending on the used proxy. Hence the present experiment basically checks the influence of a somewhat different epoch sampling. If, hypothetically, an unambiguous list of the solar 27-day extrema had been derived, this list would be used instead of the proxy data and this kind of test would be obsolete.

As expected, the results (Fig. 11) show overall a similar shape corresponding to the description in Sect. 4.3.1 with higher deviations for low and high MJO thresholds and lower deviations in the medium range. Nevertheless, there is also considerable variability among the different curves, showing that the analysis is still sensitive to the exact sampling of the epochs, although the relatively long period of 38 years is analyzed. It appears that the range of significance is somewhat different for Lyman alpha and the alternative indices; where Lyman shows significant data points between MJO strength thresholds of roughly 1 to 2, the significant range is located more between 0.3 and 1.3 for the other indices. Also this observation should, however, not be overinterpreted, since it is not evident when using the alternative numerical setup with centered epochs (Sect. 4.2, Fig. S5 in the supplement). There, the significant range is more homogenous and spans a broader range from roughly 0.3 to 1.6 for most solar proxies.

The fact that the variability introduced by a somewhat different epoch sampling propagates into the final results, suggests that it is currently safer to repeat such subtle analyses of the solar influence in tropospheric parameters with different proxies to check if the drawn conclusions are robust.





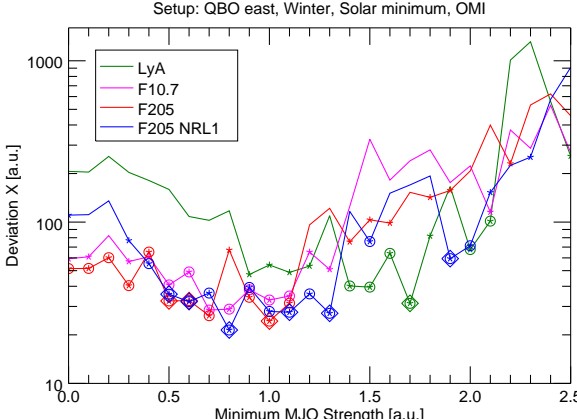

**Figure 11.** As Fig. 5, but also showing the analysis results for alternative solar proxies. The green line corresponds directly to the green line in Fig. 5. Note that this figure is shown in logarithmic scaling for an easier inspection of the comparatively slight differences.

## 4.5 Significance estimation with different Monte Carlo variants

We have estimated the significance of the individual results with a MC approach as already outlined in Sect. 4.1; for each calculation result, the analysis is repeated 1000 times with randomly modified input data. The significance is then indicated by the percentage of runs, which resulted in equal or lower deviations $X$ (stronger relationship between solar 27-day cycle and
MJO phase evolution) as the original calculation.

There is, however, a lot of freedom in the particular design of the random modification of the input data and, to our knowledge, there is no unambiguous argument for selecting a particular method. In contrast to this, the particular implementation is usually only briefly described in many studies and a comparison of the different results is difficult. In our case there is not only freedom in how the random component is implemented, but also to which of the three time series (MJO strength, MJO phase,
list of solar extrema) it is applied. Since we are analyzing here a very subtle feature, the relationship of two quasi-periodic but still variable processes of the sun-earth-system, we decided to discuss different implementations here, so that the spectrum of possible significance values becomes obvious.

The basic question for the investigation of a relationship between two quasi-periodic processes is, to which extent the random modification may influence the internal temporal behavior of both processes. On the one hand, it is exactly this internal
structure, which characterizes the inherent nature of the processes (here, e.g., the temporal evolution of the MJO) and which should not be artificially modified. On the other hand, exactly this temporal behavior has to be randomly disturbed in order to check, whether the relationship of both processes reacts to this disturbance. In other words, a random modification has to be introduced, as this is the idea of the MC technique, but is has to be kept so small that the nature of the analyzed process remains comparable. This problem is also discussed in, e.g., Davison and Hinkley (1997) or Chernick (2007) in the context of
the bootstrap method.



Since it is mostly not obvious, which idea for the random modifications meets this compromise best, we have tried different implementations, which are described in the following. The results of all implementations are compiled in Fig. 12 for the standard experiment conditions, which correspond to the example of Sect. 3: QBO easterly phase, boreal winter, MJO strength threshold 1, and solar minimum trigger. In Fig. 12, the results are ordered by a decreasing conservation of the internal structure
of the original time series. In addition to the standard MJO index OMI, we have also calculated these experiments for RMM and FMO.

To start with one extreme, it would be possible to replace one or both time series with white noise, i.e., completely not auto-correlated random data. It is intuitively clear that the application of the described analysis procedure to such a random time series would only very unlikely result in a structured pattern as seen in, e.g., Fig. 3. Hence, low probabilities of deriving
lower deviations would be found and a high significance of the original calculation would be indicated. But looking closer, this estimation would not be very conclusive, since the characteristics of the original data, which initially motivated the analysis, are not apparent anymore in the white noise random time series. Nevertheless, we have conducted two related experiments. First, we have replaced the MJO phase time series by a time series, in which the MJO phases are randomly distributed according to a uniform distribution, without any auto-correlation. Indeed, the probability to undercut the original deviation $X$ with the
random data is essentially 0% (Fig. 12, on the very right) for OMI and FMO. The probabilities for RMM are generally higher, since the relationship was weak anyway with this index (Sect. 4.4.1). But it is with about 1% still low in this case. Hence, this experiment confirms the expectation that it is unlikely to derive the sawtooth-pattern with a completely randomized MJO phase distribution. In the second approach, we have left the MJO index values untouched but have selected the dates for the solar extrema completely randomly. For this, we have selected as many out of the about 14000 possible days as have been
considered in the original analysis. Hence, the epochs are randomly distributed over the complete analyzed period and are totally independent of the actual temporal behavior of the solar proxy so that a potential temporal relation between the solar proxy and the MJO index will be broken. However, at least the temporal evolution of the MJO during the individual epochs is conserved, since the MJO index is untouched. The probability to find lower deviations with this random dataset is still below 1% for OMI (Fig. 12, second from right), which has a somewhat stronger meaning than the first experiment; it shows that the
sampling of the MJO index with epochs has to be largely systematic to reproduce the found relationship.

At the other end of extremes, one could not touch the internal structure of all time series at all. For example, the random component could be introduced by randomly selecting subsets of the originally considered epochs, which is similar to the bootstrap method. Hence, only different subsets of the same data pairs (MJO phase and solar proxy) are evaluated and it is not very surprising that this approach results in a comparatively high probability to find similar low or lower deviations. Although
we have included this result for completeness (Fig. 12, on the very left), it is not very meaningful in this context, since this experiment does not challenge the temporal relationship between both processes at all. Instead, such an analysis evaluates the influence of the particular sampling period on the result and could, e.g., be used to compute error bars for the deviations (which we have not extensively done due to a limitation of computation time). Hence, this approach is not considered further on.

As a good compromise between both extremes, we ended up with shifting the originally considered solar extrema dates a
bit (see also Sect. 4.1). Particularly, the extrema dates are shifted by a few days, which are randomly selected from a uniform





distribution between -6 d and +6 d for each solar extremum independently. Hence, this approach modifies the temporal relation between both processes, but is restrained to the effect that the evolution of the MJO is not touched at all, while the mean periodicity of the solar 27-day cycle is also conserved and only the deviations from this mean period are randomly changed. Hence, also the inherent temporal mean structure of the solar proxy is conserved when these random fluctuations are introduced.

This approach leads to the already mentioned probability of about 8% to undercut the original deviation with the random data in the present example (Fig. 12, second from left, and Fig. 4). Considering the only slight changes of the solar extrema dates (less than 6 days compared to the large MJO period variability of a few tens of days) the 8% appear remarkably low, i.e., the significance remarkably high. Formulated the other way around, shifting the solar extrema dates randomly by only a few days will already weaken the found relationship between solar variability and the MJO phase evolution in 92% of the cases. This

low probability to undercut the original deviation $X$ with this conservative approach indicates, that coincidences of variations in the solar proxy and in the MJO phase time series are not very tolerant against slight temporal changes and, hence, that a synchronization of both variations might really exist. Note that for some of the previously described experiments (Sects. 4.3 and 4.4) also significance values of better than 5% and 1% were found using this approach.

We are not aware of any unambiguous definition of the randomly generated data, but think that we have at least well

motivated the latter approach, which has been generally used as the standard method in this study. However, we do not claim that this is the only possible approach. Aside from the fact that the range for the random shifts of $\pm6$ days is an arbitrary definition, completely different approaches to generate the random data are conceivable. We have implemented two further ideas (with two variants each), which we will outline in the following. These approaches indicate even a higher significance of our results. However, as we are carrying out this subtle study as conservatively as possible, we have decided to use that

approach as the standard, which results in the lowest significance.

Both alternatives modify the MJO time series and leave the list of solar extrema dates untouched. For the first approach the continuous MJO index time series is completely shifted by a random number of days. The shifted period can be each number of days between 0 and the length of the time series. The ending period, which exceeds the original end date of the analysis after the shift, is cut and pasted in place of the now missing starting period. To our understanding, a comparable approach has also

been used in Hood (2017, 2018). For our first variant of this approach, the shift is only applied to the MJO phase evolution, whereas both phase and strength are modified similarly in the second variant. Keep in mind that phase and strength have different roles in the analysis; whereas the strength is only used as filter criterion, the phase is basically the analyzed quantity. This approach almost completely conserves the internal temporal structure of the MJO index except at the two seams. The only thing disturbed is the direct temporal day-to-day relation between solar variations and MJO variations. The disturbance is,

however, stronger than in our standard approach, since the resulting temporal difference between originally coincident features of the solar proxy and the MJO index can be many years instead of only $\pm6$ days. The results show (Fig. 12, third and forth item) a very low probability to undercut the original deviation, which is comparable to that of the totally random time series explained first. Hence, this approach would indicate a high significance, if treated as the deciding approach. The result of this approach further indicates that, in order to explain the observed relation, it is not enough to have two processes, which only act





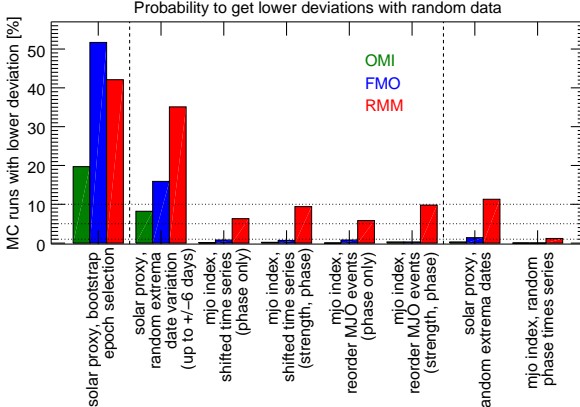

**Figure 12.** Results of the different MC implementations for the standard experiment setup (QBO easterly, boreal winter, MJO strength threshold 1, and solar minimum trigger), which corresponds to the example of Sect. 3. The 8 items are ordered according to a decreasing conservation of the internal temporal structure by the random modifications. The results are shown for the MJO indices OMI (green), FMO (blue), and RMM (red) for each implementation method. The first and the last two methods, which are separated by dashed lines, are included for completeness, but do not really characterize the significance of the experiment; whereas the first implementation does not challenge the temporal relation between solar proxy and MJO phase, the last two methods do destroy very much the internal temporal characteristics of the original time series. See Sect. 4.5 for details.

on related time scales in terms of the mean period. Instead, it seems that a closer temporal linkage on the basis of individual solar and MJO cycles could be necessary.

The second alternative is based on a random redistribution of individual MJO events (i.e., continuous periods starting with phase 1 and lasting until phase 1 is reached again), hence, the new MJO index time series are composed by randomly redis-
5   tributing MJO event pieces of the original time series. This is also either applied to the MJO phase alone or to both phase and strength. This approach also preserves the temporal structure of the MJO to a large extent, since the mean periodicity as well as the temporal behavior of the individual MJO events are not changed. But as in the previous case, the temporal relation of the solar proxy and the MJO index is strongly disturbed, since originally coincident cycles get randomly separated by possibly long periods. The results (Fig. 12, fifth and sixth item) are comparable to those of the first alternative, which also indicates that
10   the temporal relation between both processes on the basis of individual cycles seems to be important.

Note that also the result of the experiment with relaxed filter criteria (no filtering for QBO and season, see Sect. 4.3.5 and Fig. 9) based on OMI, which was not considered significant with 14.7% using the standard approach, would be significant on at least a 5% level if the latter two alternative MC approaches would be treated as decisive.



## 5 Discussion and conclusions

The MJO has been known to be a major source of tropospheric variability on the intraseasonal time scale for some decades. More recently, studies indicated that the solar 27-day cycle could introduce variability not only in the upper and middle atmosphere, but also in the troposphere. At first, this raises questions on how these sources can be unambiguously attributed to
observed variability. But even more interestingly, there have been indications that both sources are actually linked. Particularly, it has been suggested that the occurrence of strong MJO events is modulated by the solar 27-day cycle.

We have analyzed a complementary aspect, namely whether the temporal evolution of the MJO phases is potentially linked to the temporal evolution of the solar 27-day cycle. For this, we have analyzed about 38 years of MJO indices and solar proxies in combination. We have basically counted the occurrences of particular MJO phases as function of time lag after the
solar 27-day extrema. To achieve comparability between different experiments, we have developed a quantification approach based on the standard least-squares fitting routine to measure the strength of such a possible relationship. We have used this to analyze the relationship under different atmospheric conditions (state of the QBO, seasons, MJO strengths), different solar cycle triggers, and different MJO indices and solar proxies. Furthermore we have applied different implementations of a MC significance analysis and compared the results.
We have indeed found indications for a synchronization between the MJO phase evolution and the solar 27-day cycle under certain conditions, which are summarized below. Overall, the relation is such that the MJO cycles through its 8 phases within 2 solar cycles, i.e., the mean period of the MJO is twice that of the solar variation. Hence, it should be approximately 54 d, which fits well into the broad range of possible periods between 30 d and 90 d known before. The phase relation between the MJO and the solar variation is such that the MJO is predominantly either between phase 8 and 1 or between phase 4 and 5 at
the times of solar 27-day minimum. Consistently, the MJO transitions either from phase 2 to phase 3 or from phase 6 to phase 7 during solar maximum.

We have found that this relation is most pronounced during QBO easterly phases and during boreal winter, which is consistent with previous studies. The relationship can then be identified for a broad range of MJO strength thresholds between approximately 0.5 and 2.0. The upper limit is, however, probably only the artificial result of a very low number of samples
and might increase with the availability of longer datasets. For these conditions combined, the relation is surprisingly clear as shown in Fig. 3. For relaxed atmospheric filter criteria, the relation is still recognizable (e.g., Fig. 9), but is not significant anymore according to a conservative estimation.

As we have been trying to carve out a very subtle potential feature of the sun-earth-system, we have implemented not only one MC experiment as significance analysis, but several variants. The basic difference among these implementations is, to
which extent the random modifications may alter the original internal temporal structure of the time series. As our standard method, we have selected the most conservative one, i.e., the one which needs only modest modifications of the original time series and which has comparatively low significance values. In particular, we leave the MJO index time series as it is and randomly shift the solar extrema dates by up to ±6 days each. It is, however, difficult to find the only unambiguously correct variant for this particular problem, so that we have also discussed other implementations. With some of these variants, the





relation between the solar cycle and the MJO phase evolution would actually be considered significant under more diverse conditions, including the previously mentioned relaxed atmospheric filter criteria.

Although we think that the partially surprising clarity of the results justifies reporting on this topic already now, we would like to emphasize that we do not consider the relationship to be already proven. First, not in a statistical sense, since there are
many open questions left and since our analysis still suffers from a low number of samples despite the 38 analyzed years. And second, even less is clear in a causal sense, on which we have not worked so far. Even if the statistical connection is confirmed in future, it appears difficult to undoubtedly extract the exact mechanism, which would also have to explain, why the mean period of the affected process is twice that of the forcing.

One major question, which has been brought up by the present study, is, why the relationship appears so clearly when using
univariate OLR-based MJO indices like OMI and is almost not present when using multivariate indices like RMM. As OMI is known to better represent the convective center, one explanation could be that RMM simply fails to reproduce this subtle feature. However, another possibility, which cannot be neglected, is that the relationship is not really a property of the real MJO but only of its representation in OMI. Since OMI is only based on OLR data, this could mean, that we have rather analyzed a relationship between the solar 27-day cycle and OLR. Although being not our original focus, this would also be of interest, as
it would be an additional indication for the presumption that upper tropospheric parameters are influenced by solar variability. And the directly following question on the mechanism of such a potential sun-OLR relationship might refer back to the MJO. In any case, the triad solar 27-day cycle, OLR, and MJO should be subject of further studies in future.

Another major question, which could not be clearly answered by the present study, concerns the origin and the consequence of the period relation of both processes, i.e., the factor 2, which apparently connects the mean periods of the solar 27-day
cycle the MJO (54 d). This factor appears remarkable and might also support the assumption that a synchronization between the solar 27-day cycle and the MJO phase evolution really exists. However, one could also argue the other way around that this factor could be a random feature of the sun-earth system, which accidentally produces the results of the present analysis. Indeed, if one assumes that the analysis is applied to two perfect harmonic oscillations, with a factor 2 between the periods, then one would expect exactly the same sawtooth-like pattern in the results. In this case, a statistical relationship would be
found, which has no causal counterpart at all. However, this implicitly assumes that the phase between the two oscillations is constant, or at least that a particular phase relationship dominates during the analyzed period. This can, unfortunately, not be excluded based on the present analysis, but it appears at least questionable if such a dominant phase relation is plausible for such a variable phenomenon as the MJO without any synchronization mechanism. Hence, it was one aim of the conducted MC experiments to also quantify the influence of random variability in the context of these two quasi-periodic processes.
The results of different MC implementations indicated consistently that it is not sufficient to have a constant relation of the mean periods of the two processes. Instead, the results indicated that a connection on nearly a day-to-day basis is important to reproduce such a close relationship between the processes as seen in the real data. Nevertheless, such MC experiments might indicate that the probability for a pure coincidence of two processes with doubled periods is low, but they cannot disprove this possibility, so that this question remains open.



An additional outcome of this study is emphasizing the particular importance of the influences of the applied datasets and methods. Studies on this topic should be repeated with different MJO indices and also the precise meaning of applied MC analyses should be discussed. In this respect, also further efforts in method development would be valuable, which could lead to a standardization of approaches to make the results more comparable. This should also include frequency analyses, which

we have not applied here, but which could also help to understand more the appearance of the factor 2 between the periods of the solar 27-day cycle and the MJO phase evolution.

**Appendix A: Defining a measure for the strength of a relationship between the MJO phase evolution and the solar 27-day cycle**

**A1   Common measure for the goodness of fit $\chi^2$**

Commonly, analytical functions are fitted to measured data by minimizing the quantity $\chi^2 = \frac{1}{\nu} \sum_i w_i (y_i - y(x_i, a_1, ..., a_M))^2$ (e.g., Press et al., 1992). Here, the $y_i$ are the data points, $w_i$ are weights (commonly defined as $w_i = \frac{1}{\sigma_i^2}$ with the $\sigma_i^2$ being the standard deviations), $y(x_i, a_1, ..., a_M)$ is the analytical function fitted to the data points and $a_1, ..., a_M$ are the parameters, which are adjusted by the fit. The number of degrees of freedom, $\nu$, is the number of independent data points minus the number of adjusted fit parameters. In the present case $y = y(x_i, A_{St}, P_{St}, \phi_{St})$ is the sawtooth function with the amplitude $A_{St} \equiv 27\,\mathrm{d}$,

the period $P_{St} \equiv 4\,\mathrm{MJO\ phases}$, and the phase $\phi_{St}$, which is the only free parameter adjusted by the fit. The independent variable, $x$, represents the 8 MJO phases, the dependent variable, $y$, represents the time lags of maximum occurrence for each MJO phase.

After the fitting routine has determined the optimal parameter $\phi_{St}$, i.e. the one that leads to a minimal value of $\chi^2$, this value of $\chi^2$ summarizes the residual deviations between data and fit. Hence, it could be used as the sought measure of the

deviation between the data and a sawtooth function. However, two pragmatic modifications have to be applied to derive a suitable measure in the present case, which are described in the following Sections A2 and A3.

**A2   Accounting for periodicity in the fitting process**

The calculation of the individual deviations has to account for the fact that the time lags are periodic with a periodicity of 27 d. This means that, e.g., the deviation between the time lags 3 d and 23.5 d is not the comparatively large number of 20.5 d,

but only 6.5 d, as exemplified in Fig. A1. The largest deviation that can occur is therefore $27\,\mathrm{d}/2 = 13.5\,\mathrm{d}$. This has to be reflected by a modified quantity measuring the deviation between data and fit, which we define as $\chi_{per}^2 = \frac{1}{\nu} \sum_i w_i \Delta y_i^2$. The $\Delta y_i$ are initially defined to identically reproduce the original $\chi^2$, i.e, $\Delta y_i = \Delta y_{i,orig} = y_i - y(x_i, a_1, ..., a_M)$. However, after their initial calculation the values of the $\Delta y_i$ are restricted to the range between $\pm 13.5\,\mathrm{d}$ by subtracting multiples of 27 d from the $\Delta y_{i,orig}$, hence $\Delta y_i = |\Delta y_{i,orig}| - k_i \cdot 27\,\mathrm{d}$, where $k_i$ counts the multiples of 27 d to be subtracted.

Instead of the minimization of $\chi^2$ commonly used for curve fitting, we use $\chi_{per}^2$ for the present study, so that the fitting routine finds the optimal fit parameter $\phi_{St}$ under consideration of the periodicity of the fitted relationship.





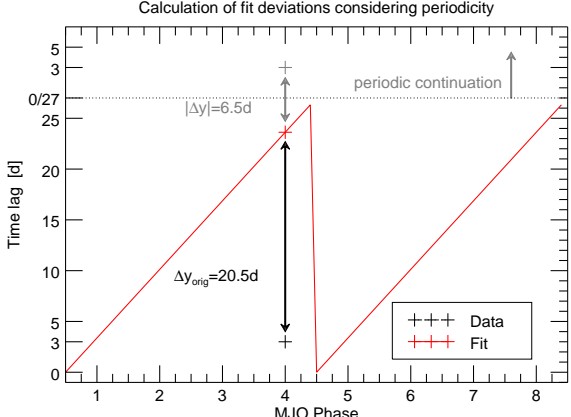

**Figure A1.** Illustration of the calculation of deviations between the data points and the fitted values. For reasons of clarity, only one data point has been included (black cross). With the conventional approach, the deviation $\Delta y_{orig}$ is simply the difference between the observed and the fitted value (red cross), which is 20.5 d in this example. However, the fitted relationship is not only periodic in the direction of the abscissa, but also in the direction of the ordinate, since possible time lags repeat themselves with a period of 27 d. The observed data can therefore be virtually periodically continued and it becomes obvious that the relevant deviation $\Delta y$ is that between the fitted data point and the newly introduced virtual data point (gray cross), which has a value of $\Delta y = -6.5$ d ($= 20.5$ d $-27$ d) in this example.

### A3 Measuring the deviation including weights

For the calculation of $\chi^2$, the individual deviations are usually weighted according to the uncertainty of the measurements $y_i$. This is adopted here also for the calculation of $\chi^2_{per}$. The weights $w_i$ are calculated as usual as the reciprocal variances of the measured data, i.e., $w_i = \frac{1}{\sigma_i^2}$ with the $\sigma_i$ being the standard deviations (The values of $\sigma_i$ are estimated with the bootstrap

method described in Sect. A4). This is, of course, a useful definition for the originally intended application of $\chi^2$ and $\chi^2_{per}$, being the quantities to be minimized during the fitting process; the relative importance of data points with a large uncertainty is reduced and the other way around.

However, such a quantity $\chi^2_{per}$ is not suitable for the intended measure of similarity between the data points and a sawtooth function; good similarity should be indicated by a small $\chi^2_{per}$ (small deviations $\Delta y_i$ between data and fit). But using this kind

of weighting, a comparatively small $\chi^2_{per}$ is also produced by large uncertainties, which is the opposite of the wanted behavior in this context.

A solution is to compute a different overall measure of the deviations $\Delta y_i$ after the fitting (which remains based on the minimization of $\chi^2_{per}$). A straightforward and pragmatic definition, which we introduce here as deviation $X$, is similar to $\chi^2_{per}$ but using reciprocal weights: $X = \frac{1}{\nu} \sum_i \frac{\Delta y_i^2}{w_i} = \frac{1}{\nu} \sum_i \sigma^2 \Delta y_i^2$. With this definition, large uncertainties lead to a higher value of

$X$, which indicates a weaker relation between the pattern of the data points and the sawtooth function. And the other way around, smaller uncertainties work in the same direction as small deviations between data and fit and lead to a small value of $X$, which indicates a stronger relation between the pattern of the data points and a sawtooth function. Hence, based on this




value the relation of the solar 27-day cycle and the MJO phase evolution can be quantified and compared between different experimental setups (e.g., different filtering or underlying datasets).

We note that this definition also has disadvantages. First, it is a somewhat arbitrary choice, particularly the power of 2, with which the standard deviations $\sigma$ contribute. It has been chosen analogously to the definition of $\chi^2$, but could have been chosen

also differently. Second, this definition combines two factors which modify the value of $X$, the deviations between data and fit and the uncertainties $\sigma$. Hence, using this measure, it cannot be distinguished whether differences of $X$ between experimental setups are dominated by the deviations or the uncertainties. The influence of this choice on our conclusions is discussed in Sect. 4.2 and results derived with an alternative choice are shown in the supplement.

## A4   Estimating the uncertainty of the found days of maximum MJO phase occurrence

Since the derived time lags of maximum MJO phase occurrence are the result of a counting process that incorporates the complete dataset, there is no possibility to directly determine the corresponding uncertainties, i.e. the statistical distribution function and its width. A well-established approach to estimate the uncertainties for such cases is the bootstrap method (e.g., Efron, 1979; Davison and Hinkley, 1997; Chernick, 2007). Basically, random samples are drawn from the original sample to generate additional virtual samples for which the complete analysis is repeated a large number of times. This results in the

distribution of possible analysis results considering random effects in the original dataset. From this distribution the uncertainty can be calculated as, e.g., the standard deviation.

In our case, we use the set of identified solar extrema dates as independent members of the original sample. From these dates we draw 1000 random samples with the same number of members (sampling with replacement) and repeat the analysis for each random sample. This results in 8 distribution functions of the time lags of maximum MJO phase occurrence, one for each

MJO phase.

Calculating the standard deviation of these distributions as uncertainty is also somewhat more complicated than usual, again due to the periodicity of the time lags (compare Sect. A2); imagine a distribution, which is centered at time lag 26 d and symmetric with wings of a few days length on both sides. Because of the periodicity the right wing will not be located around 29 d, but at time lags around 0 d to 5 d, whereas the left wing remains around 24 d. Hence, the distribution would look like

a bimodal distribution with 2 unconnected centers. The mean value would be in the middle at about 13 d and the standard deviation would represent a width, which spans the complete range from 0 d to 27 d. To overcome this problem, we shift each distribution function first, such that the maximum is located in the middle at a time lag of about 13 d and calculate the standard deviation afterwards, which is, apart from that, not affected by the shift.

*Data availability.*   The datasets used in this paper are publicly accessible.

*Competing interests.*   The authors declare that they have no conflict of interest.



*Author contributions.* CGH outlined the project, designed the method, performed the study and prepared the manuscript with substantial insight and interpretation of results provided by CvS during all previously mentioned tasks.

*Acknowledgements.* The MJO indices OMI, VPM, and FMO have been obtained online from the NOAA Earth System Research Laboratory (www.esrl.noaa.gov/psd/mjo/). The MJO index RMM has been obtained online from the Australian Bureau of Meteorology (www.bom.gov.au/climate/mjo/). Solar proxy data have been obtained online from LASP Interactive Solar Irradiance Data Center (lasp.colorado.edu/lisird/). QBO data have been obtained online from the Institute for Meteorology at Freie Universität Berlin (www.geo.fu-berlin.de/en/met/ag/strat/). This work was supported by the University of Greifswald. We acknowledge support for the article processing charge from the DFG (German Research Foundation, 393148499) and the Open Access Publication Fund of the University of Greifswald.





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
