# Peer review of "Indications for a potential synchronization between the phase evolution of the Madden-Julian oscillation and the solar 27-day cycle"

_Atmospheric Chemistry and Physics, 2018_

## Referee Comment (RC1) · Anonymous Referee #1 · 30 Nov 2018

This is a well-written and rigorous study of interdependence between Madden-Julian Oscillation (MJO) and the 27-day solar cycle. This study confirms that the relation between these phenomena is most pronounced during certain conditions: during easterly phases of Quasi-biennial Oscillation (QBO) and during boreal winter and MJO events with a strength between 0.5 and 2. On the other hand, the authors bring a rigorous and systematic analysis of all possible aspects which may influence the interpretability of the link between MJO and the 27-solar cycle. In particular, the results strongly depend on the used MJO index and indicate a possible relation between the 27-day solar

cycle and outgoing longwave radiation (OLR). Overall the manuscript has appropriate scientific quality and significance and I recommend publication in the ACP.

However, due to the extensity and focus of the study, I would appreciate an adoption of Open Science approaches to allow reproduce extensive analysis in this study (e.g. Laken, 2016). In particular, I would recommend any kind of willingness of the authors to publish the code allowing to reproduce the figures in the paper. There are multiple ways how to proceed, either to allow the access upon request or via portals allowing to assign Digital Object Identifier (DOI) to the research outputs, e.g. ZENODO. I think it could enhance the quality and reliability of this publication. In the end, this publication might be motivating for future solar-terrestrial studies.

Minor comments

In the abstract please provide explanations of the following abbreviations: RMM, OMI.

I would add a citation of the study Sukhodolov et al (2017) to the discussion starting in P3L3 or elsewhere related to the middle atmosphere. Their modeling results of temperature without a rotational component reveal that the atmosphere can produce random internal variations with periods close to 27 days even without solar rotational forcing. Furthermore, they also discuss quite extensively studies of Hood (1986) and Hood (2016).

Fig. 12 misses an explanation what horizontal lines represent.

References

Hood, L. L. (1986). Coupled stratospheric ozone and temperature responses to short-term changes in solar ultraviolet flux: An analysis of Nimbus 7 SBUV and SAMS data. Journal of Geophysical Research, 91(D4), 5264. http://doi.org/10.1029/JD091iD04p05264

Hood, L. L. (2016). Lagged Response of Tropical Tropospheric Temperature to Solar Ultraviolet Variations on Intraseasonal Timescales. Geophysical Research Letters.

http://doi.org/10.1002/2016GL068855

Laken, B. A. (2016). Can Open Science save us from a solar-driven monsoon? Journal of Space Weather and Space Climate, 6, A11. http://doi.org/10.1051/swsc/2016005

Sukhodolov, T., Rozanov, E., Ball, W. T., Peter, T., & Schmutz, W. (2017). Modeling of the middle atmosphere response to 27-day solar irradiance variability. Journal of Atmospheric and Solar-Terrestrial Physics, 152–153, 50–61. http://doi.org/10.1016/j.jastp.2016.12.004
* * *

---

## Referee Comment (RC2) · Anonymous Referee #2 · 5 Dec 2018

Review of the Article 'Indications for a potential synchronization between the phase evolution of the Madden-Julian oscillation and the solar 27-day cycle' by Christoph G. Hoffmann and Christian von Savigny

This study explored possible linkages between Madden-Julian oscillation (MJO) and the solar cycle in a 27-day timescale, using a total of 38 years data. It found some synchronising pattern under certain conditions. They identified a connection during boreal winter when QBO is in the easterly phase and MJO event strength is greater than 0.5. Such knowledge can be useful in improving predictive skills of MJO. It is an

under explored areas of research and any advancement would greatly benefit climate science community.

It is a well-written paper and followed some in-depth analyses. It discussed various limitations of this study. I would recommend its publication after major revision.

Major comments:

1. Page 2, line 2: "The amplitude of the 27-day cycle is generally smaller than that of the 11-year cycle but can be on the order of 50% of the 11-year amplitude in the UV during strong events." Probably authors would consider doing little discussions on cases if 27-day cycle matches with active or inactive phases of the 11-year cycle. Is there any connection with the 11-year cycle and how it is affecting results? Say, if the 27 day Max coincides with 11 year Min phase or 27 days Min with 11 years Max phases?

2. "We have used this to analyze the relationship under different atmospheric conditions (state of the QBO, seasons, MJO strengths), different solar cycle triggers, and different MJO indices and solar proxies." – Authors could prepare a table for Boreal winter showing QBO phase, solar 11 years Max/Min and MJO phase. Also, include the number of sample points. This will be useful to address more on the limitation part.

3. Page 2, line 35: "In addition, there are also increasing indications for an entanglement of the MJO in teleconnections and, hence, for an influence of the MJO in the extratropics" Holton Tan effect (1980) suggested some special feature in solar 11-year cycle minimum and QBO easterly, during boreal winter. Labitzke van Loon (1992) also noticed a connection in upper stratosphere polar temperature during 11-year Minimum and easterly phase. Whether polar annular modes have any connection? Discuss those works. Holton, J. R., and Tan, H.C., (1980): The influence of the equatorial quasi-biennial oscillation on the global circulation at 50 mb, J. Atmos. Sci., 37,2200-2208. Labitzke, K. and van Loon, H., (1992): On the association between the QBO and the extra-tropical stratosphere, J. Atmos. Terr. Phys., 54, 11/12, 1453–1463.

4. Labitzke, K. and van Loon, H., (1992), suggested QBO westerly solar Max in 11-year time scale also have the same influence in the upper stratosphere like solar Min/QBO Easterly in winter. Perhaps you could do little analyses using QBO westerly solar Max to check whether it is also the case for 27-day cycle.

5. Section 4.3.2: QBO phase 30 hPa or 50 hPa have any effect? Using QBO 30, boreal winter, solar Max/Ely, solar Min/Ely and solar Max/Wly are different to solar Min/Wly in solar 11-year time scale (Roy and Haigh, 2011; Camp and Tung, 2007). Those studies discussed that QBO 30hPa only indicate cold upper stratospheric pole for Solar Min/Wly in boreal winter. The rest three other combinations are warm. Is it also seen for 27 day timescale? Mention that you could verify that in future. Some analyses whether using two different QBO height give additional insight. Is it sensitive to the choice? Perhaps one plot using QBO 30 hPa?

Camp, C.D. and Tung, K. K., (2007): The influence of the solar cycle and QBO on the late-winter stratospheric polar vortex, J. Atmos. Sci., 64, 4, 1268-1283, Doi:10.1175/JAS3883.1 Roy I, Haigh JD. (2011) The Influence of solar variability and the quasi-biennial oscillation on lower atmospheric temperature and sea level pressure, Atmospheric Chemistry and Physics, pages 11679-11687, article no. 11, DOI:10.5194/acp-11-11679-2011.

Minor comments:

1. Page 2, line 17: Give references for solar 'top down' and 'bottom up' mechanism. Discuss those mechanisms which are not that clear here. 2. Fig.1,5 and 12: shorten the legend. Discuss the details within the text. 3. Page 17, Fig. 7 legend: spelling of May 4. Page 6, line 9: "Not considered extrema belong mostly to solar 11-year minimum phases"- how minimum is defined? 5. Line 32: "Univariate indices like OMI in the analysis, but can hardly be seen with multivariate indices like RMM. A weaker dependence of the results on the underlying solar proxy is also observed. Why, please give more discussion. 6. Page 4, line 2: "the range of possible MJO periods starts

close to the period of the solar 27-day cycle. And second, the mean periodicity of the MJO is with 50 to 60 days approximately twice that of the solar 27-day variability, which turns out to be of interest in the following" How many observations did you have? 7. Page 4, line 7: Future directions: "we aim at describing the statistical features of a combined inspection of both quasi-period processes as a basis for future research." Discuss whether the earlier comments using different QBO height and solar 11 year cycle gives some additional insight.

---

## Author Comment (AC1) · 24 Jan 2019

We would like to thank the two reviewers for the concise and constructive comments, which helped us to improve the manuscript significantly. The comments are repeated below followed directly by our answers.

RESPONSE TO REVIEWER 1

GENERAL COMMENT

[Figure]

COMMENT: This is a well-written and rigorous study of interdependence between Madden-Julian Oscillation (MJO) and the 27-day solar cycle. This study confirms that the relation between these phenomena is most pronounced during certain conditions: during easterly phases of Quasi-biennial Oscillation (QBO) and during boreal winter and MJO events with a strength between 0.5 and 2. On the other hand, the authors bring a rigorous and systematic analysis of all possible aspects which may influence the interpretability of the link between MJO and the 27-solar cycle. In particular, the results strongly depend on the used MJO index and indicate a possible relation between the 27-day solar cycle and outgoing longwave radiation (OLR). Overall the manuscript has appropriate scientific quality and significance and I recommend publication in the ACP. However, due to the extensity and focus of the study, I would appreciate an adoption of Open Science approaches to allow reproduce extensive analysis in this study (e.g. Laken, 2016). In particular, I would recommend any kind of willingness of the authors to publish the code allowing to reproduce the figures in the paper. There are multiple ways how to proceed, either to allow the access upon request or via portals allowing to assign Digital Object Identifier (DOI) to the research outputs, e.g. ZENODO. I think it could enhance the quality and reliability of this publication. In the end, this publication might be motivating for future solar-terrestrial studies.

ANSWER: We would like to thank reviewer 1 for this positive feedback. We understand the concern regarding the reproducibility and are open to share and discuss the code. We have included a note in the manuscript that we will make the source code available upon request. Since this is no general software package, which could be reused for other purposes, we feel that this is the most appropriate way in this case.

MINOR COMMENTS

COMMENT: In the abstract please provide explanations of the following abbreviations: RMM, OMI.

ANSWER: done

COMMENT: I would add a citation of the study Sukhodolov et al (2017) to the discussion starting in P3L3 or elsewhere related to the middle atmosphere. Their modeling results of temperature without a rotational component reveal that the atmosphere can produce random internal variations with periods close to 27 days even without solar rotational forcing. Furthermore, they also discuss quite extensively studies of Hood (1986) and Hood (2016)

ANSWER: Thank you for mentioning this study. We also think that it fits well into our introduction and have added a citation.

COMMENT: Fig. 12 misses an explanation what horizontal lines represent.

ANSWER: fixed

RESPONSE TO REVIEWER 2

MAJOR COMMENTS

COMMENT: 1. Page 2, line 2: "The amplitude of the 27-day cycle is generally smaller than that of the 11-year cycle but can be on the order of 50% of the 11-year amplitude in the UV during strong events." Probably authors would consider doing little discussions on cases if 27-day cycle matches with active or inactive phases of the 11-year cycle. Is there any connection with the 11-year cycle and how it is affecting results? Say, if the 27 day Max coincides with 11 year Min phase or 27 days Min with 11 years Max phases?

ANSWER: We agree that the influence of the solar 11-year cycle on the relation between MJO und 27-day solar cycle would be of interest. We have actually included a filtering for solar 11-year maxima and minima in our analysis. However, since it became clear that the filters for season, QBO phase, and MJO strength are essential for our analysis, the solar 11-year selection can only build on top of the these filtering steps. Unfortunately, it turned out that including an additional filter for the solar 11-year cycle reduces the number of remaining solar 27-day cycle samples down to a number (below about 10) for which no significant and reliable conclusions can be drawn anymore. Therefore, we had to postpone the work on this aspect until longer datasets are available. This also applies for some more comments below. Nevertheless, we have included a short paragraph in the manuscript (Sect. 4.3), which mentions this circumstance.

COMMENT: 2. "We have used this to analyze the relationship under different atmospheric conditions (state of the QBO, seasons, MJO strengths), different solar cycle triggers, and different MJO indices and solar proxies." – Authors could prepare a table for Boreal winter showing QBO phase, solar 11 years Max/Min and MJO phase. Also, include the number of sample points. This will be useful to address more on the limitation part.

ANSWER: Unfortunately, we do not understand very well from the comment, what exactly the table should show (what the columns should be and which information then the rows should contain). But we wonder if this comment has maybe become obsolete anyway, since we can unfortunately not address the questions regarding the solar 11-year cycle. In any case, there is Table 1 in the manuscript, which summarizes the parameters varied in the different experiments. Please note that it is not straightforward to put the experimental results themselves into a table, if this is meant, because each line in, e.g., Fig. 5 consists of 26 individual experiments (differing by the MJO strength thresholds) with different sample numbers etc. This is why we have chosen line plots for the presentation.

COMMENT: 3. Page 2, line 35: "In addition, there are also increasing indications for an entanglement of the MJO in teleconnections and, hence, for an influence of the MJO in the extratropics" Holton Tan effect (1980) suggested some special feature in solar 11-year cycle minimum and QBO easterly, during boreal winter. Labitzke van Loon (1992) also noticed a connection in upper stratosphere polar temperature during 11-year Minimum and easterly phase. Whether polar annular modes have any connection? Discuss those works

ANSWER: We agree that the interconnections of equatorial tropospheric and strato-spheric variability (MJO and QBO), polar variability, and solar variability (on different time scales) are very interesting topics. That is the reason, why we mention the possible entanglement of the MJO in atmospheric teleconnections in this sentence of the introduction. However, in all other parts our study is focused on the influences of the solar 27-day cycle (and not the 11-year cycle) on purely the MJO (equatorial, tropospheric variability). The 2 requested papers deal, instead, with teleconnections between the equatorial stratosphere (QBO) and the polar stratosphere. And whereas Holton & Tan (1980) do not discuss the solar influence, the discussion of solar influences in Labitzke & van Loon (1992) is restricted to the 11-year cycle. Hence, we think that both studies, albeit touching interesting topics in the broader context of our work, are not necessary to motivate or understand the present study. Instead, we think that enlarging on these topics would maybe even deflect the reader's attention from the main objectives of this already comprehensive manuscript. Therefore, we prefer not to discuss these papers here.

COMMENT: 4. Labitzke, K. and van Loon, H., (1992), suggested QBO westerly solar Max in 11-year time scale also have the same influence in the upper stratosphere like solar Min/QBO Easterly in winter. Perhaps you could do little analyses using QBO westerly solar Max to check whether it is also the case for 27-day cycle.

ANSWER: As mentioned above, we agree that this would be an interesting aspect to check. However, the number of remaining solar 27-day cycles is too low for reliable conclusions if we additionally filter for the solar 11-year state. This is now also mentioned in the manuscript (Sect 4.3).

COMMENT: 5. Section 4.3.2: QBO phase 30 hPa or 50 hPa have any effect? Using QBO 30, boreal winter, solar Max/Ely, solar Min/Ely and solar Max/Wly are different to solar Min/Wly in solar 11-year time scale (Roy and Haigh, 2011; Camp and Tung, 2007). Those studies discussed that QBO 30hPa only indicate cold upper stratospheric pole for Solar Min/Wly in boreal winter. The rest three other combinations are warm.

[Figure]

Is it also seen for 27 day timescale? Mention that you could verify that in future. Some analyses whether using two different QBO height give additional insight. Is it sensitive to the choice? Perhaps one plot using QBO 30 hPa?

ANSWER: We also thought about this question during the preparation of the manuscript, but did not go into the details, since the manuscript was already long. Now, we have recalculated the experiment, which evaluates the influence of the QBO, using 30hPa winds and find that this choice does qualitatively not change our conclusions. We have included the respective figure in the supplement and added a note in the manuscript. The point is here, that the periods identified as QBOE or QBOW are not shifted that much by this selection, so that the classification of most solar 27-day cycles and MJO cycles remains the same.

MINOR COMMENTS

COMMENT: 1. Page 2, line 17: Give references for solar 'top down' and 'bottom up' mechanism. Discuss those mechanisms which are not that clear here.

ANSWER: We actually cited (Hood, 2018) as the reference and already discussed the mechanisms briefly, while trying to avoid the repetition of large parts of the discussion in (Hood, 2018). However, we have rephrased the introductory sentence to make the connection to (Hood, 2018) more clear. This part reads now: "Two major classes of conceivable mechanisms are summarized by Hood (2018) and mentioned here only briefly; on the one hand the "bottom-up" mechanisms, which assume that the only slight variations of the TSI produce strong enough heating changes directly in the troposphere to generate the observed modulations in the upper troposphere. And on the other hand the "top-down" mechanisms, which consider the stratospheric effects of the stronger UV variations as starting point; via a chain of effects the stratospheric changes could result in a change of upper tropospheric static stability and with that in a change of tropospheric deep convection with implications for clouds and temperature."

COMMENT: 2. Fig.1,5 and 12: shorten the legend. Discuss the details within the text.

[Figure]

ANSWER: We have rephrased and shortened the captions. However, as these are major figures for the understanding of the study, we intentionally tried to keep the figures with the captions self-contained, which stills results in somewhat longer captions.

COMMENT: 3. Page 17, Fig. 7 legend: spelling of May

ANSWER: done

COMMENT: 4. Page 6, line 9: "Not considered extrema belong mostly to solar 11-year minimum phases"- how minimum is defined?

ANSWER: We have rephrased this sentence to make it clearer and have included quantitative thresholds.

COMMENT: 5. Line 32: "Univariate indices like OMI in the analysis, but can hardly be seen with multivariate indices like RMM. A weaker dependence of the results on the underlying solar proxy is also observed. Why, please give more discussion

ANSWER: We have added an explaining sentence on this. Since the quoted sentence belongs to the abstract, we still tried to keep it short and not to go into details here. In any case these aspects are discussed in the main part of the manuscript (Sects. 4.4.1, 4.4.2, and 5)

COMMENT: 6. Page 4, line 2: "the range of possible MJO periods starts close to the period of the solar 27-day cycle. And second, the mean periodicity of the MJO is with 50 to 60 days approximately twice that of the solar 27-day variability, which turns out to be of interest in the following" How many observations did you have?

ANSWER: We have included the information in the introduction that 38 years of MJO data are analyzed. More detailed information on how many samples are evaluated is included anyway in, e.g., Sect. 3.

COMMENT: 7. Page 4, line 7: Future directions: "we aim at describing the statistical features of a combined inspection of both quasi-period processes as a basis for future

research." Discuss whether the earlier comments using different QBO height and solar 11 year cycle gives some additional insight.

ANSWER: We have included comments related to both aspects in the discussion (Sect. 5)

REFERENCES

Holton, J. R., & Tan, H.-C. (1980). The Influence of the Equatorial Quasi-Biennial Oscillation on the Global Circulation at 50 mb. Journal of the Atmospheric Sciences, 37(10), 2200–2208. https://doi.org/10.1175/1520-0469(1980)037<2200:TIOTEQ>2.0.CO;2

Hood, L. L. (2018). Short-Term Solar Modulation of the Madden–Julian Climate Oscillation. Journal of the Atmospheric Sciences, 75(3), 857–873. https://doi.org/10.1175/JAS-D-17-0265.1

Labitzke, K., & van Loon, H. (1992). On the association between the QBO and the extratropical stratosphere. Journal of Atmospheric and Terrestrial Physics, 54(11), 1453–1463. https://doi.org/10.1016/0021-9169(92)90152-B

---

## Referee Report (RR1)

'Indications for a potential synchronization between the phase evolution of the Madden-Julian Oscillation and the solar 27-day cycle' review by Indrani Roy

It is a very well written paper. I think the authors have addressed all the issues that we mentioned. Hence, I am in favour of publishing it after minor revision.

Line 11, page 27: You mentioned, 'Note that the influence of the 11-year solar cycle on the analyzed relationship would have been also of interest. Unfortunately, the number of remaining 27-day cycles in the analyzed 38-year period was not sufficient to apply this additional filter, so that this aspect: was not further investigated here.:'

However, the main purpose was not to find the influence of 11 year solar cycle but to eliminate the solar 11 year background signal which is present on top of 27 day cycle. It is to note that solar 11 year Min and QBO Ely combination in upper stratospheric winter polar temperature showed certain specific behaviour (Labitzke and van Loon, 1992).

Also, similar argument is mentioned in Page 13, line 14 and revise those statements.

Relating to Table: It is ok to skip a Table, as long as readers know the number of data points for the signal you noted. This study is for 38 years. There are four combinations: solar Max/Wly, Max/Ely, Min/Wly and Min/Ely; that means each combination roughly has 9 years each. Moreover, the signal you noticed is only for winter. You mentioned: 'The MJO appears to cycle through its 8 phases within 2 solar 27-day cycles.' (Page 1, line 12). The mean periodicity of the MJO is with 50 to 60 days (line 7, page 4). It can give a brief idea of how many data points in those Min/Ely will have for a particular threshold/ transition of the MJO phase. That knowledge will be useful to indicate significant results.

Fig. S15: Correct the figure legend, 'It is obvious that the choice of the pressure level, at which the wind data is evaluated to define the QBO state, does qualitatively not influence the conclusions.' It would be '….conclusions for solar Minimum years- Ely phase.' Also, mention that for QBO Wly-Min, it is significant for MJO strength1.0 to 1.1 for QBO 30 hPa.

Fig. 6: Minimum MJO strength significant only between 1.4 to 2.1; but in Fig S15 it is 1.0 to 2.1. Mention in the legend if the change is due to a larger y axis scale.

Fig. S 15: Authors did a calculation using QBO Ely (30 hPa) only in Solar Min. Following Labitzke and van Loon (1992) and Camp and Tung (2007), as I mentioned in my last review, if there is any difference due to change in QBO height, it is likely to reflect in Solar Max Ely phase.

Following the same reason, Fig S3 could have been different in QBO 30 hPa for Max.

However, it is not the main focus of this analysis and can be verified later. You can mention it as a comment here to verify for future work.

Solar 'Bottom-up Mechanism': In the solar bottom-up mechanism it is better to include 'tropical Pacific'. Lately, that hypothesis is also discussed with schematic by Meehl (2008, 2009) involving tropical Pacific Nino SST, change in Walker circulation and trade wind etc.

Meehl et al. (2008): A coupled air-sea response mechanism to solar forcing in the Pacific region, J. Climate., 21(12), 2883-2897.

Meehl et al (2009): Amplifying the Pacific Climate System Response to a small 11-year Solar Cycle Forcing. Science,

---

## Author Response (AR2)

We would like to thank Indrani Roy for reviewing our manuscript a second time and the constructive comments, which helped to improve the manuscript. The comments are repeated below followed directly by our answers.

COMMENT: It is a very well written paper. I think the authors have addressed all the issues that we mentioned. Hence, I am in favour of publishing it after minor revision.

COMMENT: Line 11, page 27: You mentioned, 'Note that the influence of the 11-year solar cycle on the analyzed relationship would have been also of interest. Unfortunately, the number of remaining 27-day cycles in the analyzed 38-year period was not sufficient to apply this additional filter, so that this aspect: was not further investigated here.' However, the main purpose was not to find the influence of 11 year solar cycle but to eliminate the solar 11 year background signal which is present on top of 27 day cycle. It is to note that solar 11 year Min and QBO Ely combination in upper stratospheric winter polar temperature showed certain specific behaviour (Labitzke and van Loon, 1992). Also, similar argument is mentioned in Page 13, line 14 and revise those statements.

ANSWER: We have rewritten the mentioned sentences to point out that solar 11-year maximum conditions are slightly overrepresented in our analysis and that possible interconnections with the QBO phase have been described in the literature and should be considered in the interpretation of our results.

COMMENT: Relating to Table: It is ok to skip a Table, as long as readers know the number of data points for the signal you noted. This study is for 38 years. There are four combinations: solar Max/Wly, Max/Ely, Min/Wly and Min/Ely; that means each combination roughly has 9 years each. Moreover, the signal you noticed is only for winter. You mentioned: 'The MJO appears to cycle through its 8 phases within 2 solar 27-day cycles.' (Page 1, line 12). The mean periodicity of the MJO is with 50 to 60 days (line 7, page 4). It can give a brief idea of how many data points in those Min/Ely will have for a particular threshold/ transition of the MJO phase. That knowledge will be useful to indicate significant results.

ANSWER: We think that the reader gets in Sect. 3 already an impression of how many data points are considered; there we mention the worst case (in terms of sample size), which corresponds to the Min/Ely case you mention here. Please note, however, that we refer to solar 27-day extrema here (and not to the 11-year extrema, to which the comment seems to be related), so that the number of epochs is not halved by differentiating between solar max and min here, because each 27-day cycle can be quantified by either by the maximum or minimum date. So, here we just refer to a different way of identifying the same number of solar 27-day cycles. Apart from this, we agree that we would have to mention the sample sizes if we would treat the solar 11-year minima and maxima individually, since this would decrease the sample size. However, this is not meant in the current context.

COMMENT: Fig. S15: Correct the figure legend, 'It is obvious that the choice of the pressure level, at which the wind data is evaluated to define the QBO state, does qualitatively not influence the conclusions.' It would be '....conclusions for solar Minimum yearsEly phase.' Also, mention that for QBO Wly-Min, it is significant for MJO strength 1.0 to 1.1 for QBO 30 hPa.

ANSWER: We have extended the caption accordingly.

Fig. 6: Minimum MJO strength significant only between 1.4 to 2.1; but in Fig S15 it is 1.0 to 2.1. Mention in the legend if the change is due to a larger y axis scale.

ANSWER: The significance indications are consistent in Fig. 6 and Fig. S15. The impression that a larger range is significant in Fig. S15 comes simply from the overlapping of the significance indications for the 50hPa and the 30hPa line (the latter is missing in Fig. 6).  So there should be no need for further comments in the caption of Fig. 6.

COMMENT: Fig. S 15: Authors did a calculation using QBO Ely (30 hPa) only in Solar Min. Following Labitzke and van Loon (1992) and Camp and Tung (2007), as I mentioned in my last review, if there is any difference due to change in QBO height, it is likely to reflect in Solar Max Ely phase. Following the same reason, Fig S3 could have been different in QBO 30 hPa for Max. However, it is not the main focus of this analysis and can be verified later. You can mention it as a comment here to verify for future work.

ANSWER: As for one of the previous comments, please note that solar minimum refers here to the solar 27-day minimum according to our sentence from the beginning of the manuscript: 'When using terms like the "solar cycle", "solar maximum", etc., we always refer to the 27-day variations in this paper if not stated otherwise'. Because the papers you mention deal with the 11-year cycle, which is not directly applicable here, we think that commenting on this could lead to further misunderstandings, which is not intended by your comment. As said before (and written in the paper), we generally agree that analyzing the influence of the 11-year solar cycle on the investigated relationship is of interest, but is currently not meaningful without longer time series.

COMMENT: Solar 'Bottom-up Mechanism': In the solar bottom-up mechanism it is better to include 'tropical Pacific'. Lately, that hypothesis is also discussed with schematic by Meehl (2008, 2009) involving tropical Pacific Nino SST, change in Walker circulation and trade wind etc.

ANSWER: We have added a reference to these papers.

[revised manuscript text omitted]

**4 Alternative definition of the QBO state by 30 hPa winds**

[Figure]

Figure S15: As Fig. 6 in the manuscript, but including results for which the QBO filtering was based on the 30 hPa winds instead of the 50 hPa winds. It is obvious that the choice of the pressure level, at which the wind data is evaluated to define the QBO state, does qualitatively not influence the conclusions very much, at least in the present case for boreal winter and solar minimum. Note, however, that for two cases (MJO thresholds of 1.0 and 1.1) the relation of the solar 27-day cycle and the MJO phase evolution becomes significant also for the QBO westerly phase if the QBO filtering is based on the 30 hPa winds. This detail should be subject to future investigations.